# From Strong Plates to Weak Boundaries: Strain Localization in the Lithospheric Mantle with Low- to High-Temperature Dislocation Creep

Etienne Van Broeck<sup>1</sup>, Fanny Garel<sup>1</sup>, Catherine Thoraval<sup>1</sup>, Diane Arcay<sup>1</sup>, and D. Rhodri Davies<sup>2</sup>

Correspondence: Etienne Van Broeck (etienne.van-broeck@umontpellier.fr) and Fanny Garel (fanny.garel@umontpellier.fr)

**Abstract.** Plate-like behavior may be reproduced in numerical experiments of mantle convection if rheology combines a temperature-dependent viscosity capped with a low yield stress. Other processes limiting the mantle ductile strength have been proposed, among which low-temperature plasticity. We propose to investigate how such a rheology can promote deformation localization at the lithospheric scale. We design a 2-D thermo-mechanical model of plate extension to compare the modes of strain localization as a function of the rheological parameterization. Various experimental flow laws for olivine (diffusion creep and dislocation creep at low- and/or high- temperature) are tested, sometimes combined with a yield-stress formulation. We quantify the evolution of the deformation pattern throughout the lithosphere, and define new diagnostics to assess whether the decrease in plate viscosity is due to an increase in strain rate ('mechanical weakening') and/or in temperature ('thermal weakening'). Deformation localization leads to a new extensional plate boundary in two successive stages: (i) narrowing of the deforming zone, (ii) rapid thinning of the highly deformed lithosphere. Here, a rheology combining diffusion creep and yield stress results in a mantle weakening that is either fully mechanical, respectively fully thermal, for temperatures lower, respectively higher, than ~1300 K. Accounting for dislocation creep enables additional feedbacks between mechanical and thermal weakening within the deforming plate, thereby enhancing the efficiency of strain localization. We demonstrate the consistency of using a yield-stress approximation to cap the mantle strength for temperature below  $\sim 1000$  K. We also show that dislocation creep and yield-stress are not interchangeable for most of the lithospheric mantle (1000-1500 K), and using only the latter may overestimate the duration of plate break-up. Finally, we discuss how our results compare to natural continental rifting cases.

#### 1 Introduction

Plate tectonic theory describes the motion of rigid lithospheric plates separated by weak lubricating boundaries (Le Pichon, 1968). Observations from seismicity and strain-rate fields derived from geodetic data confirm that most intraplate regions undergo only minor deformation, while strain is concentrated within narrow, seismically active plate boundaries (Kreemer et al., 2014). Yet, broad intraplate deformation zones are also observed, for example between the India and Australia plates (e.g. Iaffaldano et al., 2018). Reconstructions of past plate configurations demonstrate that the number, size and arrangement

<sup>&</sup>lt;sup>1</sup>Géosciences Montpellier, Université de Montpellier, CNRS, Montpellier, France

<sup>&</sup>lt;sup>2</sup>Research School of Earth Sciences, The Australian National University, Canberra, ACT, Australia

of plates have evolved significantly through geological time (Morra et al., 2013). New boundaries may form either through the reactivation of pre-existing weak zones (e.g. sutures) or by the localization of initially distributed deformation, as reconstructed in incipient continental rifts (Müller et al., 2019), back-arc basins, or incipient subduction zones (Lallemand and Arcay, 2021).

Lithosphere weakening is necessary to sustain large lateral strength contrasts in the plate tectonics regime (Bercovici et al., 2015). Laboratory experiments on olivine, the dominant mineral in the lithospheric mantle, yield rheological flow laws that predict high strength at lithospheric conditions, typically ~700 MPa for temperatures below 1500 K (e.g., Kohlstedt et al., 1995). However, mantle convection simulations incorporating such 'power law' creep flow laws lead to a stagnant-lid regime when viscosity is strongly temperature-dependent (Solomatov and Moresi, 1997), implying that an additional weakening mechanism is required (Solomatov, 2004).

Over the past decades, several candidate processes of weakening at lithospheric scale have been investigated, including shear heating (Fleitout and Froidevaux, 1980; Kaus and Podladchikov, 2006; Thielmann and Kaus, 2012), anisotropic viscosity (Tommasi et al., 2009; Mameri et al., 2021), strain softening (Gueydan et al., 2014; Meyer et al., 2017; Tarayoun et al., 2019; Fuchs and Becker, 2022), and grain-size reduction with diffusion creep and/or grain-boundary sliding (Bercovici, 2003; Bercovici and Ricard, 2013; Gueydan and Précigout, 2014; Ruh et al., 2022). A common modeling approach has been to introduce a depth-dependent yield stress in combination with temperature-dependent creep, which can reproduce plate-like behavior in mantle convection models (Tackley, 2000a; Rolf et al., 2012; Coltice et al., 2017, 2019; Ulvrova et al., 2019). This parameterization is often treated as an analogue for brittle deformation. Yet, the yield stresses required to generate plate-like regimes (

#### 2 Methods: thermo-mechanical model

#### 2.1 Extension Set-Up

We construct a 2-D thermo-mechanical model of upper-mantle extension, in which lithospheric strength (viscosity) depends both on strain rate and temperature. The model domain is a 1200-km wide by 400-km deep rectangle composed entirely of mantle material (see properties, Table 1).

**Table 1.** Parameter names and values for the model set-up.

| Parameter           | Symbol            | Value        | Unit                    |  |  |  |  |
|---------------------|-------------------|--------------|-------------------------|--|--|--|--|
| Box length          | L                 | 1200         | km                      |  |  |  |  |
| Box height          | H                 | 400          | km                      |  |  |  |  |
| Surface temperature | $T_s$             | 273          | K                       |  |  |  |  |
| Mantle temperature  | $T_m$             | K            |                         |  |  |  |  |
| Material properties |                   |              |                         |  |  |  |  |
| Volumic mass        | $ ho_{ m mantle}$ | 3300         | ${\rm kg}~{\rm m}^{-3}$ |  |  |  |  |
| Thermal diffusivity | $\kappa$          | $1.10^{-6}$  | $\rm m^2\;s^{-1}$       |  |  |  |  |
| Thermal expansivity | $\alpha$          | $3. 10^{-5}$ | $K^{-1}$                |  |  |  |  |

Our focus is indeed on the weakening of the lithospheric mantle, which is a prerequisite to trigger intraplate strain localization (e.g. Gueydan and Précigout, 2014) even if the presence of a crustal layer lowers surface lithospheric strength and introduces strength contrasts (e.g. Burov and Watts, 2006).

The initial thermal state is prescribed using the half-space cooling model for a laterally uniform plate age. Boundary conditions are as follows: the top surface is fixed at  $T_s = 273$  K, the base at  $T_m = 1600$  K, and the vertical sides are thermally insulating (Fig. 1).

The initial deformation state is derived from a purely horizontal velocity field, leading to a quasi-homogeneous initial intraplate strain rate  $\dot{\varepsilon}_{t_0}$  (3.73  $10^{-16}~{\rm s}^{-1}$  for  $v_{1/2}=1~{\rm cm~yr}^{-1}$ , Fig. 2). This initial velocity field is calculated from the simulation lateral boundary conditions, where extension is imposed by prescribing symmetric, purely horizontal, depth-dependent outflow velocities (Fig. 1). This vertical profile  $v_x(z)$  is derived from a 1-D Couette shear flow with surface velocity  $v_{1/2}$  and zero velocity at the base, calculated for the depth- and temperature dependent diffusion-creep viscosity (eq. 1) of a 50-Myr old plate (Appendix A). This corresponds to an effective 'constant-velocity plate' 89-km thick, defined as the depth at which horizontal velocity deviates by more than 1 % from  $v_{1/2}$  (Garel and Thoraval, 2021). Sensitivity tests varying the plate age used to define  $v_x(z)$  confirm that this approximation does not significantly influence the results (Appendix A).

The upper boundary is a free surface (Kramer et al., 2012). At the base, a mixed condition is applied: no-slip ( $v_x = v_z = 0$ ) is imposed on the lateral segments, while a vertical inflow ( $v_x = 0$ ) is permitted across a 600 km-wide central part (Appendix B), ensuring that strain localization develops preferentially at the domain center. The central inflow,  $v_z(x)$ , evolves dynamically to

balance time-dependent surface deformation and the prescribed lateral outflow.

Figure 1. Simulation set-up in the  $400 \times 1200$  km 2-D domain. The mechanical boundary conditions (in blue) are a free top surface, and depth-dependent horizontal flow on the vertical sides, where  $v_{1/2}$  is the plate half extension rate (Appendix A). In most simulations, the box bottom is closed except over a 600-km-wide segment located between x = 300 and 900 km where vertical velocity is free and horizontal velocity set to zero. In a one simulation, this vertical inflow is imposed (see Appendix B). The thermal boundary conditions (in red) are constant temperatures at top (273 K) and bottom (1600 K), while vertical boundaries are insulating.

Figure 2. Initial conditions: temperature field from the half-space cooling model (left panel: temperature profile T(z) for a 50 Myrs-old plate) and second invariant strain rate  $\dot{\varepsilon}_{t_0}$  (right panel - here  $\sim 3.73 \ 10^{-16} \ {\rm s}^{-1}$  for a half extension rate  $v_{1/2}$  equal to 1 cm yr<sup>-1</sup>). The velocity field is depicted by black arrows, and the 900 K and 1500 K isotherms using red lines.

We first analyze different rheological parameterizations using a reference configuration with  $v_{1/2} = 1$  cm yr<sup>-1</sup> and an initial plate age of 50 Myrs (Sect. 4.1, 4.2 and 4.3). We then explore the influence of plate resistance and far-field forcing by varying plate age from 10 to 100 Myr and spreading velocity from 0.2 to 5 cm yr<sup>-1</sup> (Sect. 4.4).

# 85 2.2 Rheological parameterization

#### 2.2.1 Mechanisms of mantle deformation

Viscous deformation of the mantle is represented by combinations of (i) creep flow laws (diffusion and/or dislocation creep) and (ii) a stress-limited viscosity ('yield stress'). Creep flow laws are based on experimental and numerical studies of olivine deformation (e.g. Hirth and Kohlstedt, 1995a, b; Gouriet et al., 2019), assuming that multiple creep mechanisms may operate simultaneously (see Sect. 2.2.2).

In contrast, a yield-stress formulation has been widely adopted in geodynamical models to reproduce plate-like behavior at the surface of a convective mantle (Tackley, 2000b; Mallard et al., 2016, e.g.). Without such a cap, purely temperature-dependent rheologies tend to produce stagnant-lid regimes (Solomatov, 1995). Yield-stress parameterizations are therefore often treated as first-order proxies for pseudo-brittle deformation. Seismicity in oceanic lithosphere suggests that brittle failure extends to temperatures of 900-1000K (e.g. Engeln et al. 1986; Abercrombie et Ekström 2001; McKenzie et al. 2005). Accordingly, in few simulations, we restrict yielding to temperatures below 900-950 K, preventing it from operating in the deeper, ductile lithosphere while still allowing a pseudo-brittle response near the surface.

#### 2.2.2 Computation of mantle effective viscosity

Diffusion creep in olivine is approximated by a temperature-dependent, Newtonian viscosity:

100 
$$\eta_{\text{diff}} = A_{\text{diff}}^{-1} \cdot \exp\left(\frac{E_{\text{diff}} + P \cdot V_{\text{diff}}}{R \cdot (T + \delta T)}\right),$$
 (1)

where  $A_{\text{diff}}$  is the pre-exponential constant, corresponding to a grain size of  $\sim$ 3 mm (Garel et al., 2020),  $E_{\text{diff}}$  the activation energy,  $V_{\text{diff}}$  the activation volume, R the gas constant, P the lithostatic pressure ( $\rho gz$ ), T the temperature, and  $\delta T$  a term accounting for a 0.5 K km<sup>-1</sup> adiabatic gradient (Table 2).

Dislocation creep, which associated viscosity is non-Newtonian and strain-rate dependent, is modeled using two alternative flow laws:

1. High-temperature (HT) 'power-law' dislocation creep (Garel et al., 2020):

$$\eta_{\text{disl-HT}} = A_{\text{disl}}^{-\frac{1}{n}} \cdot \dot{\varepsilon}_{\text{disl}}^{\frac{1-n}{n}} \cdot \exp\left(\frac{E_{\text{disl}} + P \cdot V_{\text{disl}}}{n \cdot R \cdot (T + \delta T)}\right) \tag{2}$$

where  $A_{\text{disl}}$  is the pre-exponential constant,  $E_{\text{disl}}$  the activation energy,  $V_{\text{disl}}$  the activation volume, n the stress exponent and  $\dot{\varepsilon}_{\text{disl}}$  the second invariant of the strain-rate tensor associated with the dislocation creep mechanism;

2. Unified low- to high-temperature (LT-HT) dislocation creep flow law, derived from dislocation dynamics models (Gouriet et al., 2019) and calibrated against macroscale constraints on effective mantle viscosity (Garel et al., 2020):

$$\eta_{\text{disl-LT-HT}} = \frac{\sigma_{\text{disl-LT-HT}}}{2 \cdot \dot{\varepsilon}_{\text{disl}}} \tag{3}$$

with

$$\sigma_{\text{disl-LT-HT}} = A_0(T) \left( 1 + \tanh \left[ A_1(T) (\log_{10}(\dot{\varepsilon}_{\text{disl}}) - A_2(T)) \right] \right) \tag{4}$$

where  $A_0$ ,  $A_1$ ,  $A_2$  are polynomial functions of temperature (Table 2).

The bulk creep viscosity is then calculated assuming co-existing diffusion and dislocation creep:

$$\eta_{\text{creep}} = \left(\frac{1}{\eta_{\text{diff}}} + \frac{1}{\eta_{\text{disl}}}\right)^{-1} \tag{5}$$

with total strain rate partitioned between the two mechanisms:

$$\dot{\varepsilon}_{\text{creep}} = \dot{\varepsilon}_{\text{diff}} + \dot{\varepsilon}_{\text{disl}} \tag{6}$$

An iterative scheme ensures that  $\eta_{\text{disl}}$  is computed consistently from the dislocation creep strain rate rather than the total strain rate, avoiding artificial weakening (Appendix C), and allows to compute the percentage of dislocation creep in total creep deformation (Eq. C6).

**Table 2.** Variables used in the different rheological laws investigated in this study 2.2.

| Parameter                                                 | Symbol               | Value            | Unit                                |  |  |  |  |
|-----------------------------------------------------------|----------------------|------------------|-------------------------------------|--|--|--|--|
| Diffusion creep                                           |                      |                  |                                     |  |  |  |  |
| pre-exponential constant                                  | $A_{ m diff}$        | $1. 10^{-7}$     | Pa s                                |  |  |  |  |
| Activation energy                                         | $E_{ m diff}$        | 410              | $kJ \; mol^{-1}$                    |  |  |  |  |
| Activation energy                                         | $V_{ m diff}$        | $4. 10^{-6}$     | $\mathrm{m}^3 \; \mathrm{mol}^{-1}$ |  |  |  |  |
| HT d                                                      | HT dislocation creep |                  |                                     |  |  |  |  |
| pre-exponential constant                                  | $A_{ m disl}$        | $4.4 \ 10^{-17}$ | Pa s                                |  |  |  |  |
| Activation energy                                         | $E_{ m disl}$        | 540              | $kJ \; mol^{-1}$                    |  |  |  |  |
| Activation energy                                         | $V_{ m disl}$        | $6.10^{-6}$      | $\mathrm{m}^3 \; \mathrm{mol}^{-1}$ |  |  |  |  |
| stress exponent                                           | n                    | 3.5              |                                     |  |  |  |  |
| LT-HT dislocation creep (tanh)                            |                      |                  |                                     |  |  |  |  |
| $A_0 = 4.4 \ 10^8 - 5.26 \ 10^4 \ T$                      |                      |                  |                                     |  |  |  |  |
| $A_1 = 2.11 \ 10^{-2} + 1.74 \ 10^{-4} \ T$               |                      |                  |                                     |  |  |  |  |
| $A_2 = -41.8 + 4.21 \ 10^{-2} \ T - 1.14 \ 10^{-5} \ T^2$ |                      |                  |                                     |  |  |  |  |

The pseudo-brittle contribution is introduced via a non-Newtonian yield-stress viscosity:

$$\eta_{\text{yield}} = \frac{\sigma_y}{2 \cdot \dot{\varepsilon}_{\text{II}}} \tag{7}$$

where  $\sigma_y$  is a constant yield stress (200-500 MPa in different simulations) and  $\dot{\varepsilon}_{II}$  the strain rate second invariant.

The effective viscosity used in the momentum equation is taken as the minimum of creep and yield viscosities:

$$\eta = \min(\eta_{\text{creep}}, \eta_{\text{yield}}). \tag{8}$$

135

150

Numerical stability is maintained by applying cutoffs of  $10^{18}$  and  $10^{25}$  Pa s. At each location in the domain, the dominant mechanism is tracked (yield vs. creep, and within creep, diffusion vs. dislocation, for the one contributing to  $\geq 50\%$  to the total strain rate.

# 2.2.3 Strategy to investigate rheological control on lithospheric extension in numerical experiments

We investigate in various simulations different combinations of rheologies presented in the previous section, which are listed in Table 3:

- a purely Newtonian diffusion-creep viscosity (D),
- combinations of diffusion and either a HT or a LT-HT dislocation creep  $(D d_{(LT)HT})$ ,
- combinations of diffusion creep and yield-stress rheology  $(D Y_{200/500 \text{ MPa}})$ ,
- combinations of diffusion and dislocation, and yield-stress rheology  $(D d_{(LT-)HT} Y_{200/500 \text{ MPa}})$ .

The two constant yield stress values at 200 and 500 MPa are chosen as either a representative value reproducing plate tectonics in numerical models of mantle convection ((Mallard et al., 2016, e.g.), or the upper bound for the validity of the LT-HT dislocation creep (Gouriet et al., 2019; Garel et al., 2020), respectively. The simulation domain where yielding is allowed is in some simulations explicitly restricted to the expected brittle domain, here to temperatures below 900 or 950 K ( $Y_{200/500 \text{ MPa}}^{950 \text{ K}}$ ) or  $Y_{200/500 \text{ MPa}}^{900 \text{ K}}$ ).

These combinations feature different co-dependencies of the mantle viscosity on temperature and strain rate, with a transition from yield-stress to creep-dominated rheologies at different temperatures/depths. This will allow to explore different depth-distributions of deformation mechanisms: the associated feedbacks — between intraplate strain localization (increase in strain rate) and asthenosphere upwelling (increase in temperature) — are expected to lead to various scenarios and timing of lithosphere break-up under a constant extension velocity.

#### 2.3 Numerical methods

We solve the conservation equations of mass, momentum and energy for an incompressible Stokes fluid under the Boussinesq approximation using the finite-element, control-volume framework *Fluidity* (e.g., Davies et al., 2011). This approach has been widely applied in geodynamical studies, and has been extensively benchmarked against analytical and numerical reference solutions.

The governing equations are discretized on an unstructured Eulerian mesh, which is dynamically adapted throughout the simulations. Mesh refinement criteria are applied simultaneously to temperature, velocity, strain-rate, and viscosity fields, ensuring that sharp gradients such as lithospheric thermal boundaries, strain localization zones, and viscosity contrasts are adequately resolved.

165

180

The triangular element size varies between 500 m and 50 km across the domain, with the highest resolution concentrated in the lithosphere and regions of active deformation (Fig. S1 in the Supplementary material). A typical simulation employs 50,000 nodes, although this number varies depending on the minimum element size and adaptivity criteria.

#### 160 3 Post-processing diagnostics

In the following, physical fields are interpolated from the finite-element unstructured mesh onto a  $5 \times 1~\rm km^2$  regular grid to compute spatial averages or Eulerian time-derivatives. Simulations are post-processed by calculating several diagnostics (detailed hereafter) to unravel and locate the various processes leading to plate break-up: co-evolution of spatial strain localization, plate thinning and weakening, and the causes of the latter. The lithosphere is defined here as the region colder than 1500 K. This threshold is a first-order proxy, since the unique mantle material leads to a gradual transition from lithosphere to asthenosphere for temperature, deformation and velocity (Garel and Thoraval, 2021). We also focus on the deepest region of the lithospheric plate between 900 and 1500 K.

Some diagnostics are vertically-averaged over the whole lithosphere with the notation  $\langle X \rangle_{1500~\rm K}(x,t)$ , with either an arithmetic or a geometric mean depending on the X parameter span.

Finally, to allow comparison of simulations featuring different extension velocities  $v_{1/2}$ , a global average strain is estimated as  $\varepsilon(t) = \frac{2 \cdot t \cdot v_{1/2}}{L}$  where t is the time and L the domain width. All simulations are conducted up to  $\sim$ 40 % strain ( $\sim$ 500 km cumulative horizontal plate extension).

#### 3.1 Intraplate strain localization

Intraplate strain localization is first quantified using the strain rate amplification  $A_{SR}(x,z,t)$  relative to the initially-uniform intraplate strain rate  $\dot{\varepsilon}_{t_0}$  (Fig. 2), and its average (geometric mean) over the whole lithosphere thickness at each horizontal position.

To quantify how plate deformation spatially focuses through time, we track the width w(t) of the intraplate deformed zone where  $\langle A_{\rm SR} \rangle_{1500~\rm K} (x,t) > 1$ , and calculate a narrowing rate in the first stage of deformation (see Sect. 4.2). We also quantify an intraplate lateral viscosity contrast  $R_{\rm visc}(t)$ , as the ratio between the maximum vertically-averaged viscosity along x (geometric mean) and the minimum one (in the central weak zone):

$$R_{\text{visc}}(t) = \frac{\max(\langle \eta \rangle_{1500 \text{ K}}(x, t))}{\min(\langle \eta \rangle_{1500 \text{ K}}(x, t))}$$
(9)

Finally, we compute at each time the *plateness* P(t) (Tackley, 2000b; Crameri, 2018; Fuchs and Becker, 2019), defined as one minus the relative width concentrating 80 % of the total cumulative strain rate (details of the *plateness* calculation are provided in the Supplementary material and references therein).

#### 185 3.2 Plate thinning, weakening and bulk stiffness

Plate extension sets off a central thinning, and we compute the lithosphere thickness  $h_{\text{plate}}(t)$  as the minimum depth of the 1500 K isotherm along x, from which we calculate an upwelling rate in the second stage of deformation (see Sect. 4.2).

We track the increase or decrease of intraplate effective viscosity with the weakening rate W(x, z, t) (in s<sup>-1</sup>), time-derivative of the logarithm of the effective viscosity  $\eta$ :

190 
$$W(x,z,t) = -\frac{\partial \log_{10}(\eta)}{\partial t} \approx -\frac{1}{\Delta t} \log_{10}\left(\frac{\eta_{t+\Delta t}}{\eta_t}\right)$$
 (10)

 $\Delta t$  (0.05-1 Myr, adjusted to the extension velocity) is a time increment much smaller than the localization timescale. We verify that a smaller  $\Delta t$  does not change the weakening amplitude.

Positive (resp. negative) values of W indicate plate weakening (resp. hardening). We also compute the weakening rate averaged over the lithosphere  $\langle W \rangle_{1500~\rm K}(x,t)$  (geometric mean). A 10-fold decrease in viscosity in 1 Myr corresponds to a weakening rate W about 3  $10^{-14} {\rm s}^{-1}$ .

We quantify a bulk lithosphere strength in 2-D as the tectonic force F(t) (in N m<sup>-1</sup>) required to sustain constant extension velocity at the side boundaries (Bialas et al., 2010; Brune et al., 2012):

$$F(t) = H \cdot \sqrt{(S_{xx} - S_{zz})^2 + S_{xz}^2} \tag{11}$$

with 
$$S_{ij}(t) = \frac{1}{H} \int_0^H \sigma_{ij}(z,t)dz$$
 (12)

where  $S_{ij}$  is the stress tensor vertically averaged over the whole domain height H from local stress tensor  $\sigma_{ij}$  at horizontal position x = 100 km.

#### 3.3 Physical controls on plate weakening

205

Montési and Zuber (2002) addressed the issue of strain localization potential related to rheology parameterization, featuring multiple dependencies on variables being either material properties (e.g., grain size, damage) or physical states (e.g. temperature, strain rate), by defining an effective stress exponent (their eq. 2). Their approach inspired us to calculate time derivatives of viscosity, which in our study depends solely on temperature and strain rate, to discriminate and quantify thermal and mechanical contributions to weakening.

At each position of the interpolated regular grid, the time-derivative of viscosity can be decomposed as the sum of its partial derivatives, with respect to temperature and strain rate:

$$210 \quad \frac{\partial \eta(T,\dot{\varepsilon})}{\partial t} = \frac{\partial \eta}{\partial T} \frac{\partial T}{\partial t} + \frac{\partial \eta}{\partial \dot{\varepsilon}} \frac{\partial \dot{\varepsilon}}{\partial t}$$
 (13)

which is approximated, by calculating the viscosity  $\eta$  as depending on temperature T(x,z,t) and strain rate  $\dot{\varepsilon}(x,z,t)$  at each location, as:

$$\frac{\Delta \eta}{\Delta t} = \frac{\Delta \eta}{\Delta t} \Big|_{\dot{\varepsilon}} + \frac{\Delta \eta}{\Delta t} \Big|_{T} \qquad \text{with} \begin{cases} \frac{\Delta \eta}{\Delta t} \Big|_{\dot{\varepsilon}} = \frac{\eta \big( T(x, z, t + \Delta t), \dot{\varepsilon}(x, z, t) \big) - \eta \big( T(x, z, t), \dot{\varepsilon}(x, z, t) \big) \big)}{\Delta t} \\ \frac{\Delta \eta}{\Delta t} \Big|_{T} = \frac{\eta \big( T(x, z, t), \dot{\varepsilon}(x, z, t + \Delta t), \dot{\varepsilon}(x, z, t) \big) - \eta \big( T(x, z, t), \dot{\varepsilon}(x, z, t) \big)}{\Delta t} \end{cases}$$
(14)

We can now quantify the relative contributions to material weakening  $F_T$  and  $F_{SR}$ , caused respectively by the increase in temperature ('thermal' weakening) or in strain rate ('mechanical' weakening):

$$F_{\rm T} = \frac{\frac{\Delta \eta}{\Delta t} \Big|_{\dot{\varepsilon}}}{\frac{\Delta \eta}{\Delta t}} \quad \text{and} \quad F_{\rm SR} = \frac{\frac{\Delta \eta}{\Delta t} \Big|_{T}}{\frac{\Delta \eta}{\Delta t}} \quad \text{with} \quad F_{\rm T} + F_{\rm SR} = 100\%$$
 (15)

A smaller time increment  $\Delta t$  may cause short-period oscillations in the calculated  $F_{\rm T}$  and  $F_{\rm SR}$ .

#### 4 Results

#### 4.1 Diffuse vs. localized deformation during lithospheric extension

In Sect. 4.1, 4.2 and 4.3, we first perform simulations of an initially 50 Myrs-old lithosphere extended under a half spreading rate of  $v_{1/2} = 1 \text{ cm yr}^{-1}$ . The initial intraplate strain rate is quasi-homogeneous throughout the whole lithosphere:  $\dot{\varepsilon}_{t_0} \sim 3.73 \ 10^{-16} \ \text{s}^{-1}$  (Fig. 2). Depending on the rheological parameterization (Table 3), we observe two end-members of intraplate deformation pattern for the final state ( $\varepsilon \sim 40 \%$ ): either two distinct plates separated by a localized, narrow highly deformed zone, located above an asthenosphere upwelling ("localized plate boundary" Table 3), or a single plate displaying a spatially-diffuse, and relatively homogeneous deformation ("distributed deformation" in Table 3).

Two simulations representative of these end-members are shown in Fig. 3. The non-localizing simulation *ref-0* (rheology  $D-d_{\rm HT}$ , Table 2 and Fig. 3a) exhibits a long-lasting and steady intraplate deformation. This diffuse deformation mode corresponds to a slow plate thinning ( $\sim$ 0.05 cm yr<sup>-1</sup>), arising from the competition between thickening by diffusive cooling and lithosphere extension.

Simulation *ref-1* will be used as a reference for the strain localization scenario (Table 3, and Fig. 3b). Its rheology, labeled  $D - d_{LT-HT} - Y_{500 \text{ MPa}}$ , accounts for diffusion creep, low- and high-temperature dislocation creep (eq. 3) and a yield stress of 500 MPa (eq. 7). This simulation shows a progressive narrowing of the highly deformed domain around the box center, where most of the deformation is accommodated (Fig. 3b, 1-6 Myr). In a second stage, a significant asthenosphere upwelling develops under the most stretched lithosphere portion, resulting in a boundary separating two divergent plates (Fig. 3b, 12 Myr).

In some experiments, we observe an intermediate scenario in which the deformation localization is very slow. These simulations, labeled "incomplete localization" in Table 3 do not achieve fully localized deformation: the highly deformed zone remains larger than 150 km at 40 % strain, similarly to Fig 3b at 6 Myr.

250

Figure 3. Evolution of the strain rate field (second invariant) during two end-member simulations, after 1 Myr, 6 Myr and 12 Myr of extension: (a) Simulation  $D - d_{\rm HT}$  (ref-0: diffusion creep and high-temperature dislocation creep, Table 3) showing mainly a homogeneous and diffuse lithospheric thinning; (b) Simulation  $D - d_{\rm LT-HT} - Y_{500~\rm MPa}$  (ref-1, combining diffusion creep, low- and high-temperature dislocation creeps, and a yield-stress set to 500 MPa), successfully localizing intraplate deformation. The velocity field is depicted by black arrows, the 900 K and 1500 K isotherms are outlined in red.

# 4.2 A two-stage scenario of strain localization and lithosphere weakening

The spatial localization of deformation in Simulation  $\mathit{ref-I}$  is illustrated using a distance-time representation of the vertically-averaged strain rate amplification  $\langle A_{\rm SR} \rangle_{1500~\rm K}(x,t)$  (see Sect. 3.1) in Fig. 4a and of the vertically-averaged weakening rate  $\langle W \rangle_{1500~\rm K}(x,t)$  in Fig. 4b. The more deformed region ( $\langle A_{\rm SR} \rangle_{1500~\rm K} > 1$ ) narrows with time, while, simultaneously, the weakening rate increases in the vicinity of the box center. The evolution through time of the weakening rate W(z,t), computed for two vertical profiles at  $x=300~\rm km$  and at the box center  $x=600~\rm km$  are displayed in Fig. 4c and d, respectively.

We define two successive stages to describe the process of deformation localization within the mantle lithosphere:

- Stage 1, spatial focusing of the deformed zone (0-6.5 Myr): the width w of the highly deformed zone rapidly decreases with a narrowing rate up to 14 cm yr<sup>-1</sup>, while the central plate thinning rate (upwelling rate of the 1500 K isotherm) is low, around 0.3 cm yr<sup>-1</sup> (lithosphere thin from 100 to 82 km during stage 1, Fig. 5a). The weakening rate at the plate center is roughly homogeneous throughout the whole lithosphere thickness (Fig. 4d), and larger than the off-center weakening rate (Fig. 4c). As a consequence, there is a small lateral viscosity contrast ( $R_{\rm visc} 

- Stage 2, plate thinning associated with an asthenosphere upwelling (6.5-12 Myr): the central deformed zone narrows more slowly (narrowing rate around 4 cm yr<sup>-1</sup>), while the upwelling rate is almost 5 times faster than in Stage 1 (lithosphere thin from 82 to 10 km during stage 2, Fig. 5a). We define the transition time  $t_{\rm tr}$  between Stage 1 and Stage 2 from the co-evolution of w and  $h_{\rm plate}$  (see Appendix D): over about 2 Myr the upwelling progressively accelerates while the narrowing decelerates and the off-center plate hardens rapidly (Fig. 4b and c). The lateral viscosity contrast increases up to 2000 (Fig. 5b). During Stage 2, the central weakening is the highest for temperatures higher than 800 K, which corresponds to the plate region dominantly deforming by dislocation creep (Fig. 4d and Fig 6a). Outside the deforming central region, the plate viscosity is rather steady ( $W \sim 0$ ).

Figure 4. Lithospheric deformation localization and weakening in Simulation  $D-d_{\rm LT-HT}-Y_{500~MPa}$  ( $\mathit{ref-1}$ , Table 3). (a) Time-space evolution of the strain-rate amplification  $\langle A_{\rm SR} \rangle_{1500~K}(x,t)$ , computed as the vertically-averaged ratio relative to the initial homogeneous strain rate in the plate (here  $\dot{\varepsilon}_{t_0} \sim 3.73~10^{-16}~{\rm s}^{-1}$ , see Sect. 3.1). Along the black thick line, this ratio is equal to 1. (b): Time-distance plot of the geometric mean of the vertically-averaged weakening rate  $\langle W \rangle_{1500~K}(x,t)$  (Sect. 3.2). We represent only values where >60 % (resp. <40 %) of the lithosphere thickness is in weakening, averaging only the weakening values for which W>0 (resp. hardening values  $W\leq 0$ ). (c), (d) Depthtime evolution of the weakening rate W at  $x=300~{\rm km}$  (c) or 600 km (d). The pink outline corresponds to a high weakening zone where  $\frac{W}{\hat{\varepsilon}_{t_0}} > 30$ , the red lines depict the isotherms from 800 to 1500 K. Regions with different dominant deformation mechanisms are delimited by dashed dark blue contours. In all panels, the vertical dashed lines indicate the transition time,  $t_{\rm tr}$  and the plate boundary time,  $t_{\rm PB}$ .

The end of Stage 2 is defined as the time  $t_{PB}$  when the central upwelling becomes stationary (here 12 Myr, Fig. 5a), even if the plate boundary widens slightly after this time (lateral shifting of isotherms and late off-center weakening visible in Fig. 4c). The lateral viscosity contrast remains stable, with a weak plate boundary located in-between quasi-non-deforming plates (Fig. 3b). Note that there is almost no plate weakening, neither hardening, at the box center once the boundary formation is completed (Fig. 5b).

Figure 5. Simulation  $D - d_{LT-HT} - Y_{500 \, \mathrm{MPa}}$ ,  $\mathit{ref-1}$ . (a) Time evolution of the width of deformed zone w (black) and minimum lithosphere thickness (red) as defined in Sect. 3.1. An average narrowing (resp. upwelling) rate is computed over Stage 1 (resp. Stage 2). These rates can be expressed in km Myr $^{-1}$  or in cm yr $^{-1}$  (i.e. 1 cm yr $^{-1}$  = 10 km Myr $^{-1}$ ). (b) Maximum value, over the lithosphere thickness encompassed between the 273 and 1500 K isotherms, of the weakening rate at x =600 km as a function of time (eq. 10) in blue, and intraplate lateral viscosity contrast  $R_{\mathrm{visc}}$  (eq. 9) in green .

The relative contributions of temperature  $(F_T)$  and strain rate  $(F_{SR})$  to material weakening (Eq. 15) are computed over time and depth at the box center. (x = 600 km) in Figure 6a, with (i) superimposed boundaries (in dashed dark blue) delimiting the dominant deformation mechanism defined in Sect. 2.2.2, and (ii) a 'highly weakening' zone (weakening rate normalized by the initial intraplate strain rate  $\frac{W}{\frac{E}{E}} > 30$ , in pink contour Fig. 6a).

The upper part of the lithosphere is dominated by yielding, while the plate base deforms mainly in the dislocation creep regime and the underlying asthenosphere ( $T > 1500 \, \text{K}$ ) deforms mainly by diffusion creep (Fig. 6a). The temperature delimiting the (deeper) dislocation-creep and the (top) yield-dominated lithosphere, is around 800 K, increasing slightly from Stage 1 to Stage 2. The area in yielding mechanism exhibits  $F_{\text{SR}} = 100 \, \%$  and can only be weakened by an increase in strain rate. In the lower part of the lithosphere ( $800 < T < 1500 \, \text{K}$ ), the major cause of weakening smoothly switches from an increase in strain-rate during Stage 1 ( $F_{\text{SR}} > 50 \, \%$ ) to a temperature increase during Stage 2 ( $F_{\text{T}} > 90 \, \%$ ).

Figure 6. Depth-time evolution of the relative contributions to weakening of the strain rate  $(F_{SR})$  and temperature  $(F_T)$  computed at x = 600 km, for Simulations (a)  $D - d_{LT-HT} - Y_{500 \text{ MPa}}$  (ref-1); (b)  $D - d_{HT} - Y_{500 \text{ MPa}}$ ; (c) D - Y 500 MPa; and (d)  $D - d_{HT} - Y_{500 \text{ MPa}}^{950 \text{ K}}$ . Light blue lines are the isotherms from 800 to 1500 K, the high- weakening zone is outlined in pink, and the dashed dark blue lines delimits the dominant deformation mechanisms as in Fig. 4.

#### 275 4.3 Influence of the rheological parameterization on the localization scenario

When no yield-stress rheology is implemented in the top, coldest part of the lithosphere, our experiments systematically result in non-localizing simulations with laterally-distributed deformation, regardless of the creep mechanisms (Table 3), as observed for Simulation *ref-0* (Fig. 3a). This highlights the need to limit the resistance of the upper part of the lithosphere to allow plate weakening in the ductile mantle.

**Table 3.** List of numerical simulations with a half-extension rate  $v_{1/2}$  of 1 cm yr<sup>-1</sup> and an initial plate age of 50 Myr. Reference simulations for scenarios of distributed deformation 'distr. def.' (*ref-0*) or localized plate-boundary 'loc. PB' (*ref-1*) are shown in Fig 3a and b, respectively. The intermediate regime of incomplete localization 'inc. loc.' is shown in Fig. E1c and f. The characteristic times of transition  $t_{tr}$  and plate-boundary formation  $t_{PB}$  are defined in Sect. 4.2.

| Simulation name                                            | Rheological combination |            |                                   |             | Outcomes             |                      |                                 |                     |
|------------------------------------------------------------|-------------------------|------------|-----------------------------------|-------------|----------------------|----------------------|---------------------------------|---------------------|
|                                                            | diff. creep             | disl.creep | yield stress (MPa)                | final state | t <sub>tr</sub> (Ma) | t <sub>PB</sub> (Ma) | $\varepsilon_{\mathrm{tr}}$ (%) | ε <sub>PB</sub> (%) |
| D                                                          | D                       | Ø          | Ø                                 | distr. def. | -                    | -                    | -                               | -                   |
| $D-Y_{ m 200~MPa}$                                         | D                       | Ø          | 200                               | loc. PB     | 9.0                  | 19.3                 | 15.0                            | 32.2                |
| $D-Y_{ m 500~MPa}$                                         | D                       | Ø          | 500                               | loc. PB     | 12.3                 | 21.8                 | 20.5                            | 36.3                |
| $D-d_{ m HT}$ (ref-0)                                      | D                       | HT         | Ø                                 | distr. def. | -                    | -                    | -                               | -                   |
| $D-d_{HT}-Y_{200~\mathrm{MPa}}$                            | D                       | HT         | 200                               | loc. PB     | 5.0                  | 10.3                 | 8.4                             | 17.2                |
| $D-d_{ m HT}-Y_{ m 500~MPa}$                               | D                       | HT         | 500                               | loc. PB     | 6.5                  | 11.5                 | 10.9                            | 19.2                |
| $D-d_{	ext{LT-HT}}$                                        | D                       | LT-HT      | Ø                                 | distr. def. | -                    | -                    | -                               | -                   |
| $D-d_{ m LT	ext{-}HT}Y_{ m 200~MPa}$                       | D                       | LT-HT      | 200                               | loc. PB     | 5.0                  | 10.5                 | 8.4                             | 17.6                |
| $D - d_{\text{LT-HT}} - Y_{500 \text{ MPa}} $ (ref-1)      | D                       | LT-HT      | 500                               | loc. PB     | 6.5                  | 11.8                 | 10.9                            | 19.6                |
| $D-Y_{ m 200~MPa}^{ m 950~K}$                              | D                       | Ø          | 200 <sub>(for T&lt;950 K)</sub>   | inc. loc    | -                    | -                    | -                               | > 40 %              |
| $D - Y_{ m 500~MPa}^{ m 950K}$                             | D                       | Ø          | 500 <sub>(for T&lt;950 K)</sub>   | distr. def. | -                    | -                    | -                               | -                   |
| $D - d_{ m HT} Y_{ m 200~MPa}^{ m 900~K}$                  | D                       | HT         | 200 (for T < 900 K)               | distr. def. | -                    | -                    | -                               | -                   |
| $D - d_{ m HT} - Y_{ m 200~MPa}^{ m 950K}$                 | D                       | HT         | 200 (for T<950 K)                 | inc. loc.   | -                    | -                    | -                               | > 40 %              |
| $D - d_{ m HT} - Y_{ m 500~MPa}^{ m 900~K}$                | D                       | HT         | 500 (for T<900 K)                 | distr. def. | -                    | -                    | -                               | -                   |
| $D - d_{ m HT} - Y_{ m 500~MPa}^{ m 950K}$                 | D                       | HT         | 500 <sub>(for T &lt; 950 K)</sub> | loc. PB     | 12.5                 | 19.0                 | 20.8                            | 31.7                |
| $D - d_{	ext{LT-HT}} - Y_{200 \; 	ext{MPa}}^{950 	ext{K}}$ | D                       | LT-HT      | 200 (for T < 950 K)               | loc. PB     | 5.0                  | 10.5                 | 8.4                             | 17.6                |
| $D - d_{	ext{LT-HT}} - Y_{500  	ext{MPa}}^{950 	ext{K}}$   | D                       | LT-HT      | 500 <sub>(for T&lt;950 K)</sub>   | loc. PB     | 6.5                  | 11.8                 | 10.9                            | 19.6                |

We now compare the localization scenarios obtained with rheologies different from Simulation ref-1 ( $D-d_{LT-HT}-Y_{500 \text{ MPa}}$ ), either modifying the creep rheology and/or lowering the yield stress:

- yield-stress and diffusion creep only  $(D Y_{500 \text{ MPa}})$  or  $D Y_{200 \text{ MPa}}$ ,
- yield-stress, diffusion creep and high-temperature (HT) dislocation creep  $(D d_{\rm HT} Y_{500~{\rm MPa}})$ , possibly restricting the yield-stress domain to temperatures lower than 950 K  $(D d_{HT} Y_{500~{\rm MPa}}^{950~{\rm K}})$ ,
- diffusion creep and LT-HT dislocation creep with a lower yield stress of 200 MPa ( $D d_{LT-HT} Y_{200 \text{ MPa}}$ ).

The 2-stage co-evolution of the lithosphere thickness  $h_{\rm plate}$  vs. the width of the deformed zone w (Sect. 3) is similar for all simulations (Fig. 7). Deformation is well-localized for all rheologies (plateness P converges to 1 - Fig 8d), with tectonic force F strongly decreasing in Stage 2 after the transition time  $t_{\rm tr}$  (Fig. 8c). The maximum vertically-averaged weakening rate  $\langle W \rangle_{1500~\rm K} (x=600~\rm km,t)$  at the box center also peaks for all simulations around  $10^{-13}~\rm s^{-1}$  at the end of Stage 2 before  $t_{\rm PB}$  (Fig. 8f). However, strain localization is much slower in simulations  $D-Y_{200~\rm MPa}$ ,  $D-Y_{500~\rm MPa}$  and  $D-d_{HT}-Y_{500~\rm MPa}^{950~\rm K}$ 

(group 1) compared to Simulations  $D - d_{\text{LT-HT}} - Y_{500 \text{ MPa}}$ ,  $D - d_{\text{LT-HT}} - Y_{200 \text{ MPa}}$  and  $D - d_{\text{HT}} - Y_{500 \text{ MPa}}$  (group 2), as the characteristic times  $t_{\text{tr}}$  and  $t_{\text{PB}}$  are approximately doubled (Table 3). This is also evidenced by the more sluggish evolution of the deformed zone width w and the plate thickness  $h_{\text{plate}}$  at x = 600 km (Fig.8a and b for group 1).

Figure 7. Co-evolution of the minimum plate thickness  $h_{\rm plate}(t)$  as a function of the width w(t) of the most deformed zone (see Sect.3.2) for experiments in which  $v_{1/2} = 1 \text{ cm yr}^{-1}$  and the initial lithospheric age is 50 Myr. The transition time  $t_{\rm lr}$  between Stages 1 and 2 is indicated by a circle symbol (defined in Appendix D and in Fig D1), and , the localization time  $t_{PB}$  by a star symbol, with  $h_{plate}$  and w becoming steady after  $t_{PB}$ . Empty and filled symbols refer to rheologies with a yield stress set to 200 and 500 MPa, respectively.

Focusing first on Stage 1 (narrowing of spatial deformation with little plate thinning), we observe that the tectonic force F required to maintain the extension velocity (Sect. 3.2) is the largest for Simulation  $D-Y_{500~\mathrm{MPa}}$  (Fig. 8c), indicating a stronger central deformed zone, as confirmed by the lower plateness and the slower increase in lateral viscosity contrast (Fig. 8d and e). Figure 9a presents various vertical viscosity profiles and their evolution when strain rate increases ten-fold, as expected in Stage 1 where the deformation focuses (Fig. 4a). The three fastest Simulations (group 2) exhibit weaker plates than Simulation  $D-Y_{500~\mathrm{MPa}}$ , in particular between 900 and 1300 K. Interestingly, the mechanical weakening for LT-HT dislocation creep rheology (factor 3-10 in viscosity decrease in Stage 1, Fig. 9b) is comparable with the ones of HT-dislocation creep (factor 5, Fig. 9b) and of yielding (factor 10, Fig. 9b), even though the mathematical formulations differ (Eq. 3, 2 and 7). Moreover, when the top lithosphere is weaker with a yield-stress of 200 MPa compared to 500 MPa (smaller force F in Fig.8c), we note that even if deformation localization is slightly faster, viscosity constrasts remain moderate when plate boundary is formed (Fig. 8e).

315

Figure 8. Time (and strain in %) evolution of (a) width of deformed zone w, (b) plate thickness  $h_{\text{plate}}$  at x = 600 km, (c) tectonic force F, (d) plateness P, (e) lateral viscosity contrast  $R_{\text{visc}}$ , and (f) maximum weakening rate  $W_{\text{max}}$  over the whole lithosphere thickness (T < 1500 K) at x = 600 km. Simulations have the same initial plate age (50 Myr) and extension velocity ( $v_{1/2} = 1$  cm yr<sup>-1</sup>), but are computed using different rheological combinations.

Thus, localization efficiency is not solely controlled by the resistance of the pseudo-brittle part of the lithosphere, and we explain the faster localization and larger weakening rate in Stage 1 for simulations featuring dislocation creep (Fig. 8f) by an efficient non-Newtonian feedback in the ductile domain: weaker plates promote a faster deformation focusing (Fig. 4a), which causes both larger strain rates and upwelling rates (thus a temperature increase), further weakening the plate. We also note from Figure 6, which compares the physical controls on weakening at the box center, that a rheology featuring dislocation creep allows significant mechanical and thermal weakening throughout the ductile lithosphere, in particular  $F_{\rm SR} > 50$  % at the plate base (1300 < T < 1500 K) and  $F_{\rm T} > 10$  % in the mid-lithospheric layer (800 

320

This is confirmed by Simulation  $D - Y_{200 \text{ MPa}}$  (that does not feature dislocation creep), in which localization is slower than in Simulation  $D - d_{LT-HT} - Y_{500 \text{ MPa}}$ , even if the plate strength is lower, as suggested by the tectonic force in Stage 1 (30 vs. 38 TN m<sup>-1</sup>, respectively, Fig 8f). This strengthens the idea that thermal and mechanical weakenings both occurring at mid-lithospheric depths accelerate deformation localization.

Furthermore, Simulation  $D - d_{HT} - Y_{500 \, \mathrm{MPa}}^{950 \, \mathrm{K}}$  (group 1) highlights the crucial weakening of the lithosphere between 900-1300 K: if yielding is restricted to temperature lower than 950 K, then the high-viscosity thin layer at mid-lithospheric depth (Fig. 9a) is enough to hamper localization compared to Simulation  $D - d_{HT} - Y_{500 \, \mathrm{MPa}}$  (group 2 in Fig. 8).

From Stage 1 to Stage 2, we observe in the lower part of the lithosphere between 900 and 1500 K, in all simulations involving dislocation creep, a transition from strain rate-dominated weakening ( $F_{SR} > 50$  %) in Stage 1 to temperature-dominated weakening ( $F_{T} > 90$  %) in Stage 2 (Fig. 6a, b and d). On the other hand, mantle weakening in Simulation  $D - Y_{500 \text{ MPa}}$  (Fig. 6c) is either purely thermally-driven in the diffusion creep domain ( $F_{T}=100$  %) or fully controlled by strain rate increase in the yielding realm ( $F_{SR}=100$  %). The larger weakening (W > 30  $\dot{\varepsilon}_{t_0}$  pink contour Fig. 6) occurs in Stage 2 at lower temperature in the presence of dislocation creep (Fig. 6a,b,d) than in Simulation  $D - Y_{500 \text{ MPa}}$ , for which it is restricted to diffusion creep domain at T > 1300 K (Fig. 6c).

This suggests another feedback that is more efficient for rheologies featuring dislocation creep, as illustrated in Fig. 9b which presents the viscosity profile evolution during the upwelling in Stage 2 (approximated as plate age decrease from 50 to  $\sim$ 20 Myr). Plate thinning induces a larger thermal weakening at shallower depths for dislocation creep compared to diffusion creep, allowing the upwelling to weaken the bulk of the plate (not only its base), hence promoting spatial localization of deformation.

Finally, we note that a very large portion of the lithosphere undergoes yielding in Simulation  $D-Y_{500~\mathrm{MPa}}$ , limited by a temperature smoothly increasing from 1300 to 1400 K as strain rate rises during the simulation, compared to cooler temperatures of the yield-creep transition of 750-800 K for Simulation ref-1, 1000-1050 K for Simulations  $D-d_{\mathrm{HT}}-Y_{500~\mathrm{MPa}}$  and 950 K for Simulation  $D-d_{\mathrm{HT}}-Y_{500~\mathrm{MPa}}^{950~\mathrm{K}}$  (Fig. 6). Once the plate boundary is formed, almost all the lithospheric thickness is dominated by the yield stress rheology when the rheological parameterization excludes the creep flow by dislocation  $(D-Y_{500~\mathrm{MPa}})$ .

**Figure 9.** Theoretical (a) (c) viscosity profiles, and (b) (d) viscosity ratio when (a) (b) increasing the strain rate as in Stage 1 (constant plate age of 50 Myr), or (c) (d) reducing the plate thickness as in Stage 2, here approximated as a decrease in plate age (constant strain rate of  $3 \ 10^{-15} \ s^{-1}$ ), computed for the 4 rheologies described in Sect. 4.3. The equivalent viscosities for yield stresses of 200 or 500 MPa (eq. 7) are also indicated using vertical lines in panels (a) and (c).

# 4.4 Influence of initial plate age and imposed extension rate

In this section, we compare the scenarios of deformation localization for simulations featuring, on the one hand, a different initial plate age (10, 30, 50 or 100 Myrs), which alters the thermal and mechanical structure of the lithospheric mantle, and, on the other hand, a different extension velocity  $v_{1/2}$  imposed at plate sides, controlling the constant far-field extension rate (0.2, 0.5, 1, 2 or 5 cm yr<sup>-1</sup>). We consider only two rheological parameterizations for which plate age and/or extension velocity are varied:  $D - Y_{500 \text{ MPa}}$  and  $D - d_{\text{LT-HT}} - Y_{500 \text{ MPa}}$  (Table 4), since they result in clearly distinct scenarios of strain localization for the reference geodynamical set-up (initially 50-Myrs old lithosphere stretched at a half velocity  $v_{1/2}$  of 1 cm yr<sup>-1</sup> - Sect. 4.3). The corresponding diagnostics are presented in Fig. 10, 11 and in Appendix E, Fig. E1, E2.

For most combinations of initial lithospheric age and extension velocity, the overall 2-stage process of deformation localization and plate boundary formation (Sect. 4.3) remains unchanged at first order, with Stage 1 associated to narrowing of the more deformed region — with little variation in plate thickness — and Stage 2 to upwelling — with little change in the width of the deformed zone—. This is illustrated in Appendix E, Fig. E2 , which shows thickness vs. width for different ages and velocities and indicates that their co-evolution does not depend much on the rheology parameterizations, nor significantly on the extension velocity, with plate age setting the initial plate thickness. As in the reference set-up, we always observe a faster strain localization in Stage 1 for a rheology including dislocation creep compared to the sole combination of yield and diffusion creep (Fig. 10a). The mean plate viscosity  $\langle \eta \rangle_{1500~\text{K}}$  at x=600~km is 5-10 times lower for rheology  $D-d_{\text{LT-HT}}-Y_{500~\text{MPa}}$  compared to  $D-Y_{500~\text{MPa}}$ , with greater lateral viscosity contrast between the weak boundary and the stiff plate interior (Fig. 10e and f).

For two experiments at very low half-extension rate  $(0.2 \text{ cm yr}^{-1})$ , the formation of a new plate boundary is not achieved at 40 % strain (for rheology  $D - Y_{500 \text{ MPa}}$ , 50 or 100 Myrs-old plates, 0.2 cm yr<sup>-1</sup>, Table 4). We will comment on this result below.

The initial plate age does not alter much the duration of strain localization: the strain necessary to complete plate boundary formation  $\varepsilon_{PB}$  remains encompassed between 15 and 20 % (resp. 30-36 %) for rheology  $D - d_{LT-HT} - Y_{500 \text{ MPa}}$  (resp.  $D - Y_{500 \text{ MPa}}$ ), as presented in Table 4. By contrast, we find that, whatever the rheology, increasing the extension velocity, hence intraplate strain rate, results in (i) lower viscosity in the deformed zone at  $t_{tr}$ , (ii) faster localization during Stage 1 (decrease in  $\varepsilon_{tr}$  and increase in narrowing rate) Fig. 10a, c and e). Indeed, non-Newtonian viscosities (yielding and dislocation creep (Eq. 3 and 7) lessen the effective viscosity at high strain rates, thus accelerating deformation localization in Stage 1.

When the extension velocity is larger, plate thinning in Stage 2 is also faster (Fig. 10d), with larger weakening amplitude and stronger mechanical component compared to simulations with small  $v_{1/2}$  (Fig. 11). A second-order effect is the shift to higher temperature of the deformation mechanism transition from yield stress to dislocation creep dominant and dislocation to diffusion creep dominant for  $D - d_{\text{LT-HT}} - Y_{500 \text{ MPa}}$  rheology (Fig. 11), or from yield stress to diffusion creep for  $D - Y_{500 \text{ MPa}}$ 

**Table 4.** List of simulations investigating different lithospheric ages at spreading onset and half-spreading rates  $v_{1/2}$ . Three rheological combinations are tested (first column). The last two columns list the average strain obtained at the characteristic times of transition  $t_{\rm tr}$  and plate-boundary  $t_{\rm PB}$ , respectively (Sect. 3). The initial strain rate in the plate ( $\dot{\varepsilon}_{t_0}$ ) is homogeneous equal to: 7.47  $10^{-17}$ , 1.87  $10^{-16}$ , 3.73  $10^{-16}$ , 7.47  $10^{-16}$  and 1.87  $10^{-15}$  s<sup>-1</sup> respectively for  $v_{1/2} = 0.2$ , 0.5, 1, 2 and 5 cm yr<sup>-1</sup>.

| Set-uj                                | p parameterizatio               | n                 |             |                       | Outcomes              |                            |                                    |
|---------------------------------------|---------------------------------|-------------------|-------------|-----------------------|-----------------------|----------------------------|------------------------------------|
| rheology                              | $v_{1/2}  ({\rm cm \ yr}^{-1})$ | initial age (Myr) | final state | t <sub>tr</sub> (Myr) | t <sub>PB</sub> (Myr) | $arepsilon(t_{ m tr})(\%)$ | $\varepsilon(t_{\mathrm{PB}})$ (%) |
|                                       | 0.2                             | 50                | inc. loc    | -                     | -                     | -                          | > 40 %                             |
|                                       | 0.2                             | 100               | inc. loc    | -                     | -                     | -                          | > 40 %                             |
|                                       | 0.5                             | 50                | loc. PB     | 27.5                  | 46.0                  | 22.9                       | 38.4                               |
|                                       | 1                               | 10                | loc. PB     | 13.8                  | 21.3                  | 23.0                       | 35.5                               |
|                                       | 1                               | 30                | loc. PB     | 12.5                  | 22.0                  | 20.9                       | 36.7                               |
| $D-Y_{500~\mathrm{MPa}}$              | 1                               | 50                | loc. PB     | 12.3                  | 21.8                  | 20.5                       | 36.3                               |
|                                       | 1                               | 100               | loc. PB     | 12.3                  | 20.5                  | 20.5                       | 34.2                               |
|                                       | 2                               | 50                | loc. PB     | 4.8                   | 9.5                   | 15.8                       | 31.7                               |
|                                       | 5                               | 10                | loc. PB     | 1.9                   | 3.5                   | 15.5                       | 29.2                               |
|                                       | 5                               | 50                | loc. PB     | 1.4                   | 3.7                   | 11.7                       | 30.9                               |
|                                       | 5                               | 100               | loc. PB     | 1.4                   | 3.6                   | 11.3                       | 29.7                               |
|                                       | 0.2                             | 10                | loc. PB     | 46.1                  | 96.1                  | 15.4                       | 32.1                               |
|                                       | 0.2                             | 50                | loc. PB     | 44.1                  | 95.0                  | 14.7                       | 31.7                               |
|                                       | 0.2                             | 100               | loc. PB     | 45.0                  | 99.0                  | 15.0                       | 33.0                               |
|                                       | 0.5                             | 50                | loc. PB     | 14.0                  | 26.0                  | 11.7                       | 21.7                               |
|                                       | 1                               | 10                | loc. PB     | 5.8                   | 9.5                   | 9.6                        | 15.9                               |
|                                       | 1                               | 30                | loc. PB     | 6.3                   | 10.5                  | 10.4                       | 17.5                               |
| $D-d_{	ext{LT-HT}}-Y_{	ext{500 MPa}}$ | 1                               | 50                | loc. PB     | 6.5                   | 11.8                  | 10.9                       | 19.6                               |
|                                       | 1                               | 100               | loc. PB     | 6.8                   | 12.8                  | 11.3                       | 21.3                               |
|                                       | 2                               | 50                | loc. PB     | 2.8                   | 5.0                   | 10.0                       | 16.7                               |
|                                       | 5                               | 10                | loc. PB     | 1.1                   | 1.7                   | 9.2                        | 13.8                               |
|                                       | 5                               | 50                | loc. PB     | 1.2                   | 2.0                   | 9.6                        | 16.3                               |
|                                       | 5                               | 100               | loc. PB     | 1.2                   | 2.2                   | 10.0                       | 18.3                               |
|                                       | 0.2                             | 10                | distr. def  | -                     | -                     | -                          | -                                  |
|                                       | 0.2                             | 100               | distr. def  | -                     | -                     | -                          | -                                  |
| $D-d_{ m HT}$                         | 1                               | 50                | distr. def  | -                     | -                     | -                          | -                                  |
|                                       | 5                               | 10                | distr. def  | -                     | -                     | -                          | -                                  |
|                                       | 5                               | 100               | distr. def  | -                     | -                     | -                          | -                                  |

rheology (Fig. E1). Thus, a higher proportion of the plate is prone to mechanical weakening, even in Stage 2, which accelerates the thinning of the lithosphere (see also deeper transition from diffusion creep to yield for higher strain rate in Fig. 9a).

385

Figure 10. Diagnostics as a function of half-extension velocity  $v_{1/2}$  for two different rheologies,  $D - Y_{500 \,\mathrm{MPa}}$  (red) and  $D - d_{\mathrm{LT-HT}} - Y_{500 \,\mathrm{MPa}}$  (blue). The symbols correspond to initial plates ages ranging from 10 to 100 Myrs. (a) Strain  $\varepsilon(t_{\mathrm{tr}})$ , as a proxy for duration of Stage 1. (b) Difference  $\varepsilon(t_{\mathrm{PB}}) - \varepsilon(t_{\mathrm{tr}})$  as a proxy for duration of Stage 2. (c) Narrowing rate during Stage 1. (d) Upwelling rate during Stage 2. (e) Effective viscosity vertically-averaged over the lithosphere thickness ( $T < 1500 \,\mathrm{K}$ ) at  $t_{\mathrm{tr}}$  and at the horizontal position  $x = 600 \,\mathrm{km}$ . (f) Maximum lateral viscosity contrast ( $R_{\mathrm{visc}}$  max.) during the two stages, the maximum is reach around  $t_{\mathrm{PB}}$ . (g) Tectonic force F at time  $t_{\mathrm{tr}}$ . In all panels, solid lines simply join the markers corresponding to the initial lithospheric age of 50 Myrs, used as a reference age.

As  $v_{1/2}$  increases from 0.2 to 5 cm yr $^{-1}$ , strain at the transition time  $\varepsilon_{\rm tr}$  is roughly halved for rheology  $D-Y_{500~\rm MPa}$ . In contrast, for rheology  $D-d_{\rm LT-HT}-Y_{500~\rm MPa}$ ,  $\varepsilon_{\rm tr}$  decreases more modestly from 15 % ( $v_{1/2}$ =0.2 cm yr $^{-1}$ ) to a plateau around 10 % for  $v_{1/2} \ge 1$  cm yr $^{-1}$  (Fig. 10a). This behavior is consistent with the isocontour of weakening normalized to initial intraplate deformation rate ( $\frac{W}{\dot{\varepsilon}_{t_0}} > 30$  - pink contour on Fig. 11) which shows that, for all initial plate ages, a larger intraplate region is affected as extension velocity increases from 0.2 to 1 cm yr $^{-1}$ . However, the overall weakening pattern remains similar for  $v_{1/2}$  of 1 or 5 cm yr $^{-1}$ . This plateau in 'localization efficiency' is further reflected in the stagnation of the lateral viscosity contrast for both rheologies as extension rates increase from 2 to 5 cm yr $^{-1}$  (Fig. 10f).

In simulations with slow extension (0.2 cm yr<sup>-1</sup>), or young plates (10 Myrs) undergoing a mild extension speed ( $v_{1/2} \lesssim 1$  cm yr<sup>-1</sup>), the lithosphere first undergo a moderate to strong thickening during Stage 1 (Fig. 11 and E1). This thickening rate is  $\sim$ 0.1 cm yr<sup>-1</sup>, and is higher for younger plates: thermal diffusion can overcome extension for low  $v_{1/2}$  and young plates. In these experiments, significant lithospheric weakening occurs later in Stage 1, with purely mechanical weakening ( $F_{SR}$ =100 %) also present within the dislocation creep domain at temperatures up to 1300 K).

Figure 11. Depth-time evolution of the relative contributions to weakening  $F_{\rm T}$  and  $F_{\rm SR}$  computed at the horizontal position x=600 km where deformation is maximum for rheology  $D-d_{\rm LT-HT}-Y_{\rm 500~MPa}$ . Panels are displayed with rows varying by initial plate age and columns by half-extension velocity  $v_{1/2}$ . The central panel corresponds to the reference setup (initial plate age of 50 Myrs, half-extension rate of 1 cm yr<sup>-1</sup>). The background colorbar is the same as Fig. 6, light blue lines are the isotherms from 800 to 1500 K, the high- weakening zone is outlined in pink, and the dashed dark blue lines delimits the dominant deformation mechanisms as in Fig. 4 and Fig. 6.

At such low intraplate strain rates, the region dominated by yield stress exhibit limited weakening, resulting in very slow lithosphere thinning. In the extreme cases of a  $0.2~{\rm cm~yr^{-1}}$  half extension rate with rheology  $D-Y_{500~{\rm MPa}}$ , only a thin basal portion of the lithosphere can be weakened by the temperature increase. This weakened base is minor compared to the 4-5 times thicker yielding-dominated lithosphere (Fig. E1). Consequently, plate thinning progresses too slowly to achieve localization before the total strain reaches 40 %. Even for simulations that do achieve localization, for  $D-d_{\rm LT-HT}-Y_{500~{\rm MPa}}$  or  $D-Y_{500~{\rm MPa}}$  rheology, the lateral viscosity contrast at  $t_{\rm PB}$  remains below 200 for a  $0.2~{\rm cm~yr^{-1}}$  half extension, indicating that the plate boundary remains rather stiff and thick – at least  $\sim$ 50-km thick for the rheology  $D-d_{\rm LT-HT}-Y_{500~{\rm MPa}}$  regardless of the initial plate age.

#### 5 Discussion

405

#### 400 5.1 New diagnostics for weakening physical controls

Beyond the specific rheological comparisons, this study highlights the viscosity decrease ('weakening') during strain localization (Fig. 4), and introduces new diagnostics to quantify the physical origin of lithospheric weakening (Sect. 3.3). These new diagnostics go beyond the classical assessment of strain localization using plateness (Tackley, 2000b) or shear zone relative width (Montési, 2013). We do not quantify in detail the growth rate of an instability (Kaus and Podladchikov, 2006) nor calculate *a priori* effective rheological properties (Montési and Zuber, 2002; Schmalholz and Fletcher, 2011). Rather, the new diagnostics  $F_T$  and  $F_{SR}$  apply to self-consistently evolving temperature and strain rate fields to quantify their separate contributions to weakening in a geodynamical setting.

Combined with more classical geometrical diagnostics of strain localization and plate thinning (Sect. 3.1 and 3.2), this framework offers several advantages. First, it disentangles the respective roles of mechanical and thermal feedbacks, enabling us to track how weakening evolves during different stages of localization (Fig. 4 and 6) and how it is influenced by plate age and extension rate (Fig. 11). Second, it allows us to analyze where in the lithosphere either thermal weakening, associated with both dislocation and diffusion creeps, or mechanical weakening, associated with either dislocation creep or yield-stress rheology, dominates (Fig. 6). Therefore, we can compare simulations with any rheological parameterization, including a combination of several creep mechanisms depending on the same physical state. Here, it highlights why dislocation creep, enabling both thermal and mechanical weakening, exerts such a strong influence on lithospheric weakening in the range 900-1200 K.

These diagnostics open new opportunities for application beyond our study on lithospheric extension with temperature and strain rate-dependent viscosities. They could be used to assess the role of dislocation creep in subduction zone initiation (e.g., Gurnis et al., 2004; Billen and Hirth, 2005; Ueda et al., 2008; Zhong and Li, 2019; Zhang et al., 2021; Arcay et al., 2023), to test weakening feedbacks above mantle plumes (Thoraval et al., 2006; Ballmer et al., 2009; Agrusta et al., 2015), or to investigate co-dependent fields such as grain size (e.g., Bercovici and Karato, 2002; Fuchs and Becker, 2019; Dannberg et al., 2025), which itself evolves dynamically with temperature and strain rate. More broadly, such diagnostics could be used to

distinguish between weakening driven either by inherited properties (e.g. mechanical anisotropy, damage or smaller grain size) or by evolving thermo-mechanical fields, to analyze how damaged lithospheric zones may further promote plate break-up in their surroundings (e.g. Tommasi et al., 2009; Bercovici and Ricard, 2014; Fuchs and Becker, 2022).

# 5.2 Weakening of lithospheric mantle by yielding vs. dislocation creep

Our simulations show that weakening in the ductile domain partly controls the scenario of strain localization, with feedback efficiency depending on the rheological parameterization.

First, dislocation creep and yield-stress are not interchangeable at moderate-to-high temperatures (1000-1500 K): they both promote lithospheric weakening, but in different ways. Simulations without dislocation creep (e.g.,  $D-Y_{500~MPa}$  or  $D-Y_{200~MPa}$ ) localize much more slowly than those that include it (e.g.  $D-d_{HT}-Y_{500~MPa}$  or  $D-d_{HT}-Y_{200~MPa}$ ). In the absence of (any) dislocation creep (rheologies D-Y), lithosphere weakening at the base of the plate cannot entail from the strain rate increase in Stage 1, since the diffusion creep viscosity is Newtonian ( $F_{SR}=0$  below 1300 K in Fig 6c, and no mechanical weakening in Fig. 9b). Furthermore, the absence of dislocation creep also prevents the colder heart of the mantle lithosphere (800 

compatible with the transition between a 500-MPa yield-stress and either HT-dislocation creep ( $\sim$ 1000 K, Fig. 6b), or LT-HT dislocation creep (800-900 K, Fig. 6a), but not with the transition between yield and diffusion creep for rheology  $D-Y_{500 \text{ MPa}}$  (1300 K, Fig. 6c).

This thermal transition between the pseudo-brittle yielding and creeps deformation mechanisms depends both on the strain rate and on the rheological parameterizations for creep and yield stress. The presence of a mantle layer dominated by a pseudo-brittle yield stress-rheology will also depend on the geological setting. For example, a yield-stress rheology in the mantle might not be needed below a 30-km thick continental crust when a low-temperature dislocation creep is implemented, except for very old plates (Fig. 11g,h,i). In contrast, we expect a pseudo-brittle mantle underneath a 10-km thick oceanic crust (Fig. 11).

Third, the low-to-high-temperature dislocation creep (Eq. 3) leads, at temperatures lower than 1000 K, to much lower differential stresses  $\sigma_{\text{disl-LT-HT}}$  (Eq. 4) compared to high-temperature dislocation creep (e.g. Fig. 9). This is in agreement with other formulations of olivine low-temperature plasticity (Evans and Goetze, 1979; Karato, 2008; Mei et al., 2010; Kawazoe et al., 2009; Korenaga and Karato, 2008) as compiled by Jain et al. (2017).

Our study demonstrates that low- and high-temperature dislocation creep is a realistic mechanism able to induce significant weakening in the 'cold heart' of the lithospheric mantle, in between ~800-1300 K. In the absence of dislocation creep, no thermal (resp. mechanical) weakening is possible in the cold region (resp. the bottom) of the lithosphere below 1100 K (resp. above 1300 K). Thus, models with a rheology combining only diffusion creep and yield stress (e.g. Coltice et al., 2019) may underestimate localization efficiency and over-predict the duration of continental break-up. In addition, low-temperature creep also explains deformed slab morphologies in the transition zone (Čí žková et al., 2002; Garel et al., 2014).

#### 5.3 Limitations of the present study

The simulations presented in this study feature a unique (mantle) material, and do not explicitly model the presence of crustal layers with their own vertical strength variations (e.g. Burov, 2011a). We discuss further in Sect. 5.4 how our results can relate to natural cases of lithospheric rifting and spreading.

In addition to dislocation creep, mantle strength in the range 800-1300 K may also be controlled by the accumulation of damage/strain or the reduction in grain size (e.g. Precigout et al., 2007; Bercovici et al., 2015; Fuchs and Becker, 2021; Li and Gurnis, 2024). The interplay - and relative importance - of low-temperature dislocation creep relative to such strain-weakening effects will depend on the respective parameterizations for rheology and properties evolution, and is beyond the scope of the present study. Our simulations exhibit lithosphere weakening and strain localization even though no shear heating is implemented. The latter has been shown to enhance localization through thermal weakening (Fleitout and Froidevaux, 1980; Kaus and Podladchikov, 2006; Thielmann and Kaus, 2012; Arcay et al., 2023), and we expect that, in our set-up, this would lead to an enhanced thermal weakening during Stage 1 and 2 where dissipation is expected to be large (increasing strain rates in the narrowing deformed zone), especially for regions deforming under dislocation creep.

Finally, we explore in this study only a controlled lateral extension on a 2-D domain. Including high-temperature dislocation creep in whole-mantle convection simulations may also increase plateness in mobile-lid regime, or decrease plate mobility in other cases, even possibly leading to stagnant-lid convection (Arnould et al., 2023). Indeed, in addition to lithosphere stiffness, the rheological parameterization affects lithosphere-asthenosphere viscosity contrast and mechanical decoupling, thus altering mantle flow wavelength and overlying plates' deformation and mobility (Arnould et al., 2023; Semple and Lenardic, 2021), with yet another feedback expected between plate velocities and asthenosphere weakening through dislocation creep (Patočka et al., 2024).

#### 5.4 Comparison with natural cases and models of rifting

To assess the applicability of our modeling results to geological settings, an interesting test is to evaluate whether the modeled forces, strain localization scenarios and its duration can be compared with extensional plate boundary formation in numerical modeling and in nature.

Numerical models of continental rifting typically involve a 30-40 km thick crustal layer, and a 100-120 km thick lithosphere (Huismans and Beaumont, 2007; Brune et al., 2014; Gueydan and Précigout, 2014; Chenin et al., 2018; Tetreault and Buiter, 2018), which corresponds to a plate older than 50 Myrs. The simulated mode of rifting is mainly controlled by the lower crustal rheology. A very weak lower crust (i.e., < 50 MPa at depth > 20 km, e.g. using rheologies of wet quartzite, feldspar or wet feldspar, respectively Gleason and Tullis, 1995; Rybacki and Dresen, 2004; Rybacki et al., 2006) promotes mechanical decoupling between the crust and mantle, leading to a wide-rifting style (e.g., Gueydan et al., 2008; Tetreault and Buiter, 2018; Pérez-Gussinyé and Liu, 2022) This style of rifting exhibits an upwelling while deformation is still distributed in the crust, which is a different scenario from the two-stage evolution highlighted in our results.

In contrast, when a stronger lower crust (i.e., < 300 MPa at depth > 20 km, e.g. using dry quartzite or mafic granulite, respectively Ranalli and Murphy, 1987; Wilks and Carter, 1990) is present and promotes mechanical coupling between the crust and mantle, rifting starts with a broadly distributed deformation, both in the crust and mantle. The deformed zone then smoothly narrows, with crustal faulting and associated mantle shearing. This stage is followed by crustal and lithosphere thinning, associated to a rapid asthenosphere upwelling (e.g. Huismans and Beaumont, 2007; Brune et al., 2014; Gueydan and Précigout, 2014; Chenin et al., 2018).

Thus, the narrow-rifting style is comparable to our two-stage localization. We propose that the yield stress parameterization in our models approximates a mechanical coupling between a strong crust and the underlying lithospheric mantle (Fig. 9).

The absence of crustal layers in our models may also explain why the tectonic forces during Stage 1, in the simulations including dislocation creep (Fig 8c), range between 20 and 50 TN m<sup>-1</sup>, which nears or exceeds the upper bound of available tectonic forces, e.g. rifting extension ( $\sim 1$  TN m<sup>-1</sup>, Brune et al., 2023), ridge push ( $\sim 2-3$  TN m<sup>-1</sup>, Parsons and Richter, 1980), or slab pull, likely to drive plate kinematics in the vicinity of a subduction zone ( $\sim 10-50$  TN m<sup>-1</sup>, e.g., Conrad and Lithgow-Bertelloni, 2002; Toth and Gurnis, 1998; Schellart, 2004; Wu et al., 2008).

Moreover, our simulations exhibit slower localization than other numerical models of 'narrow' continental rifting under constant-velocity boundary conditions. In these models with velocities  $v_{1/2}$  of 0.4 to 0.5 cm yr<sup>-1</sup>, the transition time  $t_{tr}$  can be estimated around 5 Myrs, and the plate-boundary formation  $t_{PB}$  between 9 and 16 Myrs (Brune et al., 2014; Gueydan and Précigout, 2014; Chenin et al., 2018), compared to respectively 14 and 26 Myr in the model  $D - d_{LT-HT} - Y_{500 \, MPa}$  (Table 3). Even lower velocities ( $v_{1/2} = 0.15 \, \text{cm yr}^{-1}$ ) exhibit plate break-up in ~40 Myrs (Huismans and Beaumont, 2007), compared to 95 Myr in our simulation with  $v_{1/2}$ =0.2 cm yr<sup>-1</sup> (Fig. 11). This likely comes from the very strong yielding layer at 500 MPa extending up to the surface in our simulations, while crustal resistance is expected to be rather low at the surface, increasing with depth up to ~300 MPa. Moreover, the presence of rheological discontinuity between lower crust and mantle is expected to enhance strain localization in the mantle (e.g., Allemand and Brun, 1991; Gueydan et al., 2008). Besides, we do not include any strain softening in our study as opposed to most rifting models (e.g. Huismans and Beaumont, 2003, 2007; Brune et al., 2013; Tetreault and Buiter, 2018; Heckenbach et al., 2021), which@@ is known to enhance strain localization in the mantle (Frederiksen and Braun, 2001).

The compilation by Brune et al. (2017) suggests that the Iberian Newfoundland margin and the central South Atlantic rift segment have recorded an early phase of distributed deformation through crustal faulting, followed by a necking phase and rapid strain localization that ultimately lead to mantle exhumation and breakup. However, geological observations cannot precisely reconstruct the detailed evolution of lithosphere thinning relative to strain localization -e.g. Iberia-Newfoundland margin Pérez-Gussinyé et al. (2006); Pérez-Gussinyé (2013); Mohn et al. (2015); Tucholke et al. (2006, 2007) and South AtlanticChaboureau et al. (2013)-, and cannot disentangle a 'narrowing' Stage 1 from a subsequent 'upwelling' Stage 2 as described in our results (Sect. 4.3). Similarly, in our simulations the narrowing of deformed zone overlaps with the incipient upwelling (Fig.5 & 7).

Nevertheless, some natural continental rift systems (for example : North East, Central North and South Atlantic rift, North America - Iberia rift or Australia-Antartica rift) seem to present an early stage of deformation, at (total) extension rates slower than 1 cm yr $^{-1}$ , lasting  $\sim$ 25-50 Myr, followed by an acceleration of extension over only 2 to 10 Myr (Brune et al., 2016). This abrupt plate acceleration was reproduced in numerical models of rifting using constant tectonic boundary forces at domain sides (Brune et al., 2016), interpreted as a consequence of 'the rapid decrease of rift strength'. It was also described by Ulvrova et al. (2019) in 2-D spherical annulus models of mantle convection, proposed as the result of 'a positive feedback loop between extension velocity and rift strength loss'. In our models with constant-velocity imposed at domain sides, we observe a roughly constant tectonic force F during Stage 1 (Fig 8), since the lithospheric weakening remains low (Fig 4d). The rapid decrease in F occurs during Stage 2 (Fig. 8), associated to intense lithosphere thermal weakening (Fig.4-b & -d, Fig. 6). One may speculate that, assuming constant tectonic forces instead of constant velocity boundaries, the decrease in plate resistance at time  $t_{\rm tr}$  would result in an increase in extension velocity, as predicted by rifting models and reconstructed scenarios in natural cases (Brune et al., 2016).

Thus, rifting acceleration in nature could represent a geodynamic tipping point associated with a positive feedback between

560 asthenospheric upwelling and lithosphere weakening, as long as far field forces remain stable over durations long enough.

Back-arc basins, such as the Lau and South Fiji basins (Tonga subduction zone), the West Philippines, Parace Vela and Shikoku, and Izu-Bonin-Marianas basins (IBM subduction zone) and the Japan sea (Japan-Kuril subduction zone), are relatively short-living, lasting less than ~15 to 20 Myr, including both the formation of the new extensional plate boundaries and the time window of active spreading (Sdrolias and Müller, 2006; Deschamps and Lallemand, 2002; Jolivet et al., 1994; Martínez et al., 1995; Martinez and Taylor, 2002; Taylor et al., 1996; Okino et al., 1998). These studies suggest that spreading generally initiates along or near the volcanic arc, which might represent a pre-existing localized zone of weakness at lithospheric scale. It is possible that Stage 1 would be strongly reduced during the formation of back-arc basins since deformation localization might be rapidly achieved thanks to a structural heterogeneity rather than through progressive narrowing as in our models.

#### 6 Conclusions

Our study demonstrates how lithospheric weakening under extension emerges from the interplay of rheology, physical feed-backs, and tectonic forcing. The simulations reveal a robust two-stage process: an initial mechanically driven narrowing of deformation zones, followed by thermally accelerated weakening as asthenospheric upwelling intensifies. Low- to high-temperature dislocation creep provides the most realistic mechanism for weakening the mantle lithosphere across the ductile domain. While yield-stress rheologies can mimic some aspects of this behavior, particularly at low temperatures, they cannot capture the feedbacks associated with high-temperature dislocation creep. As a result, models restricted to diffusion creep plus yield stress tend to overestimate the duration of strain localization and the timescale of plate break-up.

This study introduces new diagnostics that decompose viscosity changes into contributions from temperature and strain rate. By disentangling the thermal and mechanical drivers of weakening, this framework clarifies why rheologies including dislocation creep localize efficiently and complements existing measures such as effective stress exponents, viscosity contrasts, and shear-zone widths. These diagnostics provide a transferable tool to evaluate weakening feedbacks in other geodynamical contexts, from subduction initiation to plume—lithosphere interaction and grain-size—sensitive rheologies.

Together, these findings establish dislocation creep as critical to achieve realistic plate-boundary formation, demonstrate the robustness of the two-stage weakening scenario across a wide range of plate ages and spreading velocities, and provide a new quantitative framework for systematically comparing lithospheric weakening processes across tectonic environments.

590

605

610

# Appendix A: Outflow at lateral boundaries

We compute a velocity profile  $u_x(z)$  for a 1-D Couette flow

- with the half extension velocity  $v_{1/2}$  at the top and a null velocity at the bottom,
- for a vertical viscosity profile following the diffusion creep rheology (associated to a temperature profile from the half-space cooling model for a 50 Myr-old plate). The 400 km-thick layer corresponding to the domain thickness H is discretized into 400 constant-viscosity layers.

Incompressible Navier-Stokes equations at steady-state for this 1-D horizontal Couette flow simplify to:

$$\frac{d^2 u_x}{dz^2} = 0 \tag{A1}$$

595 Within each layer, the shear stress is constant and relates to the local viscosity  $(\eta_i)$  as:

$$\tau_{xz} = -\eta_i \frac{d \, u_x}{d \, z} \tag{A2}$$

Continuity of horizontal velocity and shear stress is enforced at each layer interface. Together with the imposed boundary conditions, this leads to a well-posed linear system which solution provides the horizontal velocity profile  $u_x(z)$ .

The outflow velocity profile imposed at the domain sides in our simulations  $v_x(z)$  is then defined as:

$$v_x(z) = \begin{cases} v_{1/2}(1 - \frac{z}{z_{\text{LAB}}}) + V(z_{\text{LAB}})\frac{z}{z_{\text{LAB}}} & \text{for } z \le z_{\text{LAB}} \\ V(z) & \text{for } z > z_{\text{LAB}}, \end{cases}$$
(A3)

where V(z) is a 10th-order polynomial interpolation of the Couette velocity profile  $u_x(z)$  below  $z_{\rm LAB}$  (polynomial coefficients listed in Table S1 in the Supplement).  $z_{\rm LAB}$  is the depth of 'constant-velocity' plate, defined following Garel and Thoraval (2021) as the depth where the horizontal velocity drops below 99 % of the surface extension velocity  $v_{1/2}$  ( $z_{\rm LAB}=89$  km for a 50 Myrs-old plate).

We perform sensitivity tests to assess the impact of using a single plate age (50 Myrs) to compute the outflow horizontal velocity profile, for all initial thermal plate ages in the simulations (10 to 100 Myrs). Comparisons between simulations using a 50 Myr plate age and those using 10 Myr (resp. 100 Myr) for the Couette flow calculation, in setups with an initial plate age of 10 Myr (resp. 100 Myr), showed only minor differences. The overall scenario and feedbacks remained similar, with only slight variations in timing, whether expressed in Myr or in terms of accumulated strain (Fig. S3 in the Supplement).

620

# Appendix B: Inflow at the bottom boundary

At the bottom boundary, a purely vertical inflow is allowed over a region of width  $L_{in}$ =600 km centered at the middle of the domain, i.e., at x = L/2 = 600 km. Outside this region, a no-slip condition is imposed with  $v_x(x) = v_z(x) = 0$ .

To ensure mass conservation, the total vertical inflow through this open bottom section should balance the total horizontal outflow through the side boundaries, which is twice the integral of the horizontal velocity profile  $v_x(z)$ :

$$\int_{\frac{L}{2} - \frac{L_{in}}{2}}^{\frac{L}{2} + \frac{L_{in}}{2}} v_z(x) \, dx = 2 \int_0^H v_x(z) \, dz \tag{B1}$$

Within the central open section, the vertical velocity  $v_z(x)$  is left free to adjust, with the central inflow peaking at  $\sim 1$  cm yr<sup>-1</sup> for  $v_{1/2} = 1$  cm yr<sup>-1</sup> (Fig. S4 in the Supplement). The lateral restriction of the inflow ensures that strain localization develops preferentially at the domain center, without the need for an initial heterogeneity in the lithosphere — e.g. a weak seed (Huismans and Beaumont, 2003, 2007; Tetreault and Buiter, 2018), or a thermal perturbation (Brune et al., 2012; Heckenbach et al., 2021).

In one simulation — with the same rheology, initial plate age and lateral spreading as in Simulation *ref-1* (Sect. 4.2) — the bottom inflow is enhanced, rather than letting  $v_z$  adjust to the flow dynamics, by imposing along the bottom boundary

625 
$$v_z(x) = a\left(x - \frac{L_f}{2}\right)^2 + b \tag{B2}$$

with a given maximum vertical velocity  $v_{\rm max}$ =6 cm yr $^{-1}$  at the center. The coefficients a and b, and the width of the open bottom section  $L_f$  (here  $\sim$  130 km, Fig. S4) are derived from the mass conservation (Eq. B1) assuming  $v_z(x=L/2)=v_{\rm max}$ .

As a result, the strain localization is faster in the presence of this enhanced vertical inflow (Fig B1a), with a shorter Stage 1 (transition time  $t_{\rm tr} = 4.8$  vs 6.5 Myr, associated to a faster narrowing rate) and a comparable Stage 2 duration (localization time  $t_{\rm PB} = 9.8$  vs. 11.8 Myr, i.e., less than 0.5 Myr difference when removing the shift in Stage 1 duration). The enhanced asthenosphere upwelling results in a higher intraplate strain rate at the box center, hence a slightly higher mechanical weakening ( $F_{\rm SR} \geq 60$  % in the dislocation creep domain in Stage 1 compared to  $F_{\rm SR} \geq 50$  % in Simulation ref-1, Fig. B1b).

This simulation may mimic the lithosphere deformation ahead of a plume arrival: increasing the vertical mantle inflow leads to stronger weakening and faster strain localization, even without introducing a thermal anomaly or melt weakening.

Figure B1. Evolution of (a) vertically-averaged strain rate amplification and (b) relative contributions to weakening  $F_T$  and  $F_{SR}$  computed at x = 600 km for the simulation with an imposed vertical inflow peaking at 6 cm yr<sup>-1</sup>. The transition time  $t_{tr}$  is 4.8 Myr, while the formation of the plate boundary is achieved at  $t_{PB}$ =9.8 Myr. Dashed contour refers to Simulation *ref-1* (free vertical bottom inflow) which outline in panel (a) the strain-rate amplification equal to 1 (in black) and in panel (b)  $F_{SR}$  > 60 % and  $F_T$  

This iterative procedure is initialized by computing the dislocation creep viscosity from the total creep strain rate,  $\eta_{\rm disl}^0 = \eta_{\rm disl}(\dot{\varepsilon}_{\rm creep},T)$  and the creep viscosity following eq. C1 as  $\eta_{\rm creep}^0 = \left(\frac{1}{\eta_{\rm diff}} + \frac{1}{\eta_{\rm disl}^0}\right)^{-1}$  with  $\eta_{\rm diff} = \eta_{\rm diff}(T,z)$ . For the next iteration steps  $(j \geq 1)$ , the assumption of a single stress (eq. C3) is used to modify the dislocation creep strain rate as:

655 
$$\dot{\varepsilon}_{\text{disl}}^j = \dot{\varepsilon}_{\text{creep}} \frac{\eta_{\text{creep}}^{j-1}}{\eta_{\text{disl}}^{j-1}}$$
 (C4)

which is used to compute the dislocation creep viscosity as  $\eta_{\rm disl}^j = \eta_{\rm disl}(\dot{\varepsilon}_{\rm disl}^j,T)$ . We then calculate a new creep viscosity as:

$$\eta_{\text{creep}}^{j} = \left(\frac{1}{\eta_{\text{diff}}} + \frac{1}{\eta_{\text{disl}}^{j}}\right)^{-1}$$
(C5)

Finally, the amount of strain rate accommodated by dislocation creep is the ratio between  $\dot{\varepsilon}_{\text{disl}}$  and  $\dot{\varepsilon}_{\text{creep}} = \dot{\varepsilon}_{\text{diff}} + \dot{\varepsilon}_{\text{disl}}$  which, because of the condition of a single deviatoric stress, also corresponds to the ratio between the total creep viscosity and the dislocation creep viscosity:

$$\frac{\dot{\varepsilon}_{\text{disl}}}{\dot{\varepsilon}_{\text{creep}}} = \frac{\dot{\varepsilon}_{\text{disl}}}{\dot{\varepsilon}_{\text{diff}} + \dot{\varepsilon}_{\text{disl}}} = 1 - \frac{\dot{\varepsilon}_{\text{diff}}}{\dot{\varepsilon}_{\text{creep}}} = \frac{\eta_{\text{creep}}}{\eta_{\text{disl}}}$$
(C6)

670

### Appendix D: Transition time $t_{\rm tr}$ calculation

For every "localized" simulations (loc. PB, Table 3, Table 4), we estimate the transition time  $(t_{tr})$  from the plate thickness  $(h_{plate}(t))$  evolving as a function of the width of the highly deformed zone (w(t)). The transition time  $t_{tr}$  is defined as the point furthest from the line connecting the lithospheric structures at  $t_0$  and  $t_{PB}$ , to the evolution curve  $h_{plate}(t) - w(t)$  (Fig.D1). This point corresponds to the knee in the curve and corresponds to a transition between the first stage of lithospheric spreading during which deformation narrowing dominates (strong decrease in w while  $h_{plate}$  only slightly lessens) and the second stage, in which the main development of the upwelling of the bottom of the lithosphere is on the contrary enhanced (moderate w reduction but strong lithospheric thinning and  $h_{plate}$  reduction). Such a methodology used to define a threshold has been previously proposed, for example, to estimate the threshold in grain orientation spread to discriminate between recrystallized and relict grains of quartz from a given distribution of intracristalline lattice distortion in each grain (e.g. Cross et al., 2017).

Figure D1. Definition of the transition time  $t_{\rm tr}$  illustrated by the reference experiment, Simulation ref-1 (rheology  $D-d_{\rm LT-HT}-Y_{\rm 500~MPa}$ ). The transition time  $t_{\rm tr}$  is determined as the time when the distance between the co-evolution of the deformed-zone width w and the plate thickness  $h_{plate}$  (blue line), and the straight line linking the end-member states at t=0 and  $t=t_{PB}$  (black line) is maximum.

### **Appendix E: Additional Figures**

Additional Figure E1 displays the results obtained for the  $D-Y_{500\,\mathrm{MPa}}$  rheology and for various initial plate ages and half-extension rates (Sect. 4.4). In particular, Fig. E1c and f illustrate a scenario of incomplete localization (Sect. 4.1, Table 4), where localization is not achieved at the end of the simulation when strain reaches 40 %.

Figure E1. Depth-time evolution of the relative contributions to weakening  $F_{\rm T}$  and  $F_{\rm SR}$  computed at the horizontal position x=600 km where deformation is maximum for rheology  $D-Y_{\rm 500~MPa}$ . Panels are displayed with rows varying by initial plate age and columns by half-extension velocity  $v_{1/2}$ . The central panel corresponds to the reference setup (initial plate age of 50 Myrs, half-extension rate of 1 cm yr<sup>-1</sup>). Light blue lines are the isotherms from 800 to 1500 K, the high- weakening zone is outlined in pink, and the dashed dark blue line delimits the dominant deformation mechanisms as in Fig. 4, Fig. 6 and Fig. 11.

Additional Figure E2 provides a global illustration of the two-stage evolution in simulations for various initial plate ages and half-extension velocities (Sect. 4.4). The curves for the same initial plate age are almost superimposed, except for low velocity  $v_{1/2} = 0.2$  cm yr<sup>-1</sup> which exhibits thickening during the early Stage 1. The 'incomplete localization' scenario of the

simulation with rheology  $D - Y_{500 \,\mathrm{MPa}}$  for an initial 50 Myrs-old plate and  $v_{1/2} = 0.2 \,\mathrm{cm}\,\mathrm{yr}^{-1}$  (Sect. 4.1, Table 4) is illustrated by the pink curve in Fig. E2b: at the end of the simulation (strain of 40 %), the plate is still thicker than 700 km and the deformed zone wider than 150 km.

Figure E2. Co-evolution of the highly deforming width w(t) and of the plate thickness  $h_{plate}$  (approximating the evolution of the deforming lithosphere morphology), for two rheological combinations, when the initial plate age and the half-spreading velocity are varied. (a) Rheology:  $D - d_{\rm LT-HT} - Y_{\rm 500~MPa}$ . (b) Rheolog:  $D - Y_{\rm 500~MPa}$ .

# Code and Data availability

The Fluidity computational modeling framework, including source code and documentation, is available from https://fluidityproject. github.io (release 4.1.19, Kramer et al. (2021)). Post-processing codes were developed using Python (3.11.11) and are available upon request from the corresponding author, with few Figures using scientific color maps of Fabio Crameri. Simulation data supporting the findings of this study are available at https://zenodo.org/records/17606932, including raw files necessary to reproduce the main numerical experiments. Additional simulation outputs not included in the repository are available upon request from the corresponding author.

# **Author contribution**

EVB, FG, and RD designed the numerical experiments; EVB, FG, CT, and DA developed new post-processing diagnostics; EVB developed the codes for simulation analysis; EVB and CT produced the figures; all authors discussed the results and contributed to the writing of the paper; FG supervised the project and coordinated the research.

# 695 Competing interests

The authors declare that they have no conflict of interest.

# Acknowledgements

The authors thank S. Demouchy, A. Tommasi, N. Coltice, and M. Arnould for fruitful discussions. The numerical simulations were performed using the *Fluidity* code and realized with the support of the HPC Platform MESO@LR, funded by the Occitanie/Pyrénnées-Méditerranée Region, Montpellier Mediterranean Metropole and the University of Montpellier.

# Financial support

700

This research was funded by the French National Research Agency (ANR) through the RheoBreak project (grant no. ANR–21-CE49-0009) led by Fanny Garel.

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
