# Peer review of "From Strong Plates to Weak Boundaries: Strain Localization in the Lithospheric Mantle with Low- to High-Temperature Dislocation Creep"

_EGUsphere, 2025_

## Referee Comment (RC1)

**Review of "From Strong Plates to Weak Boundaries": Strain Localization in the Lithospheric Mantle with Low- to High-Temperature Dislocation Creep" by Etienne Van Broeck, Fanny Garel, Catherine Thoraval, Diane Arcay, and D. Rhodri Davies**

**Antonio Manjón-Cabeza Córdoba**

Dear editor,

this technical paper presents a series of numerical experiments to address the role of rheology on deformation localization during extensional tectonics. While several detailed models of extension and rifting exist, evaluation of localization and adequate rheological mechanism is mostly absent in modern literature. In addition, they add the role of low-temperature plasticity, a continuation of their own work (Gouriet 2019; Garel 2020)

They show that considering multiple rheological mechanisms, in particular non-linear rheologies, is important to reproduce localization processes as we imagine them on Earth. They could specify a bit more on whether this was ever called into question, or what are the consequences of ignoring one rheology or the other, not just for their models, but for the wider community that will be forced to take simplifying assumptions regardless.

The manuscript is quite technical but the methodology is well explained. I have little to say there. Because Solid Earth has a wide focus - inasmuch as the deep Earth can be called a wide field – the article would benefit for a slightly more careful contextualization (e.g. what advances or new evidence motivate this study? How does this work compare with other studies? What are the implications for other geophysicists/geologists? etc). But this is not a criticism of the science of the manuscript itself.

Overall, I have few important comments to make and the majority of them are oriented to increase the reader interest on the manuscript rather than a criticism of the science of the manuscript. As a matter of taste, I find that they describe too many details of their results; these are simplified far-from-reality numerical experiments, and describing too many details may have little relevance. Perhaps summarizing a bit would improve the manuscript (i.e. in some sections there are some details that would be obvious to most people, like with faster extension comes faster thinning).

First you can find the main comments to be addressed by the authors, below you can find a series of minor suggestions that the authors are free to address or not, or even ignore during review/reply process.

**Formatting**

I'm sure this will be fixed at production stage, but please check spacing between paragraphs.

Extra paragraph spaces: 268, 294, 305, 324, 351, 361, 364, 371, 408, 441,460,481,502,519, 537, 556, 561. At least those, I think. If these are intentional, then correct paragraphs without break (e.g. 465-466; 469-470...)

**Abstract**

I hate to make this comment (I myself am not a great writer), but please revise the language of the abstract. Some constructions are confusing. I have tried to suggest some changes, but I am not a native speaker and at least some of the coauthors are. After reading the whole manuscript I have noticed this is something very punctual, and that the quality of the writing is good overall. Probably the abstract was written last, and a quick revision would make the manuscript much more attractive.

**Introduction**

One of the highlights of the manuscript is a new way to measure localization. Beyond the weird wording (we were inspired by -> we build on?), it seems that the main analysis of localization goes through the new diagnostics described in the manuscript. And yet, the justification pertaining to this new diagnostic is quite lacking. What was wrong with previous ways to measure localization? What new information does this diagnostic bring? This is something that can be addressed in 2-3 lines as a motivation in the introduction section, and would give way to another couple of sentences in the discussion defending the diagnostic role on improving our knowledge of localization (rather than justify the diagnostics *a posteriori*).

Please, check my comment below on the references.

**Methods**

The methods are very thorough in the description of the model, which I appreciate.

I would appreciate a plot showing that the Arrhenius description of high-T dislocation creep converges at high T with the tan-description of low-T-high-T dislocation creep. Otherwise, we could be comparing apples with oranges. But this can be left for the supplementary material or one of the appendixes (note to editor: the equations are described well enough that the reader can plot this him/herself).

Lines beyond 215: I think this definition would be clear(-ish) to most geodynamicists but there are better ways to write this. As it stands, one could argue that if $F_t$ and $F_r$ add to 100% (which they don't necessarily do unless normalized, which it is not explicit in the formulas), then hardening on one of the

parameters (Ft or Fr) should result in a negative percentage in one of the parameters and a value greater than 100% in the other (white areas in figures 6, 11?). Then the reader may or may not reach the (incorrect or correct) conclusion that the authors use the absolute values of the partial derivatives in equation 14 (not explicit) and in equation 15, so that the values are necessarily constrained to 0-100, but then being unable to distinguish between weakening or hardening (not to mention that Delta eta/ Delta t would not be representative of deta / dt). I can't help but feel I am overthinking this, but the authors could help the reader by being a bit more explicit/explanatory.

**Results**

The results are very thoroughly described. Little to add here. If anything, I'd suggest to simplify the results. Obviously, the authors can present a lot of good data, but I wonder how much of it is needed to justify the conclusions of the paper. To some extent, some of the data could be a bit of an overkill (See number figures/panels in Figure 8, 10, etc). In general I would summarize section 4.3 (it was difficult to read, in addition, although to be fair that could be my own limitations).

**Discussion**

Limitations

The authors find that dislocation creep and yield stress are not interchangeable, which implies that both, non-linear rheology and a yield stress are necessary to reproduce rifting/ridges scenarios. This, however obvious it may seem, is an important result that greatly aids the geodynamics community (I am opinionated in this regard). But it turns out that the only existing physical process close to our models' yield stress is brittle failure. Our continuum models are inherently limited to volumetric (as opposed to planar) failure, and therefore can only reproduce this form of deformation approximately. While I don't believe so, it could be argued that considering planar failure would change the results. In addition, the classical approach to yield stress does not converge at the resolution of our models (although the relevant associated parameters may indeed converge), and more advanced approaches may give different results (Duretz et al., 2020; 2023). These considerations are arguably more important than the limitations highlighted in the discussion section at the preprint stage, and I suggest adding something along these lines. Nonetheless, I'd like to point out that these minor details do not disqualify the calculations of the paper.

The cases with yield stress include a constant yield stress with depth. I believe this is a major limitation that should be discussed. As with the previous paragraph, the only physical mechanisms similar to the yield stress used here is the brittle failure, whose friction coefficient implies an increase in the strength with depth. Note that

I am not suggesting that the model setting is not adequate (this simplification may aid in the disentangling of the different mechanisms/weakenings), but the limitation itself needs to be discussed.

Implications

While I know this is not the objective of the authors, it would be of benefit to the paper to include a couple of paragraphs on how the inclusion (or exclusion) of certain rheologies may influence more realistic models. I am always puzzled when authors claim to match real systems even when ignoring/neglecting key pieces of the puzzle. In this case, I am sure low-T plasticity has been ignored in most of the literature. How does low-T plasticity influence wide or narrow rifting? What may classical models be underestimating/overestimating? This would be much more interesting to the rather than "[...] the yield stress parameterization in our models approximates a mechanical coupling [...]" of specific models, which seems to me more a sentence to justify the model setting than implications (Note: IMO we should always design the models with an experiment/model in mind, not run the model and then find a suitable application; this may be too idealistic, the authors may disregard).

As it stands, a lot of the implications sound more like a review of current knowledge of rifting, with too loose of a relation with this manuscript's experiments (e.g. in lines 538-545, whether this work would have predicted one thing or the other, this paragraph could be written exactly the same regardless of the results).

**Conclusions**

I disagree with the use of the word demonstrate. While this word does not necessarily need to mean the same as in a math paper, I don't think these numerical experiments can demonstrate anything, not without discarding other theories/mechanisms/processes, as with any experimental work. What the calculations here described show is that within the assumptions of the model, we can discard one or two rheologies, at least to explain by themselves some form of localization. That does not amount to a demonstration to me. Just to clarify, the paper does demonstrate that complex rheology is indeed a good way to localize deformation, but this was never into question (as far as I know) and it does not demonstrate that other mechanisms not here considered are not viable. To sum up, without considering other mechanisms/processes this study cannot demonstrate "[...] how lithospheric weakening under extension emerges [...]", just one of several possibilities.

Sorry for the lengthy paragraph, what I meant is the following: consider changing "demonstrates" for "shows" or a similar, less-charged word.

The sentence "models restricted to diffusion creep plus yield stress tend to overestimate the duration of strain localization and the timescale of plate break-up" is perhaps the most important conclusion of this work, or at the very least the most impactful. Consider discussing a bit more about this in the discussion (e.g., could the authors find some works where they think this timescale has been overestimated? if so, consider discussing them, but I understand if the authors consider this too conflictive and therefore not suitable for this article).

**References**

The cited references are a bit unbalanced.

I understand the affinity argument for citations, but I feel the citations are a bit too close to home. Although I am partial to succinct manuscripts, and the current citation number stands at ~120, I feel like the manuscript could be a bit more balanced in terms of research groups (and/or nationality of the cited researchers). I don't want to be overly disingenuous, though, and, to me, featuring more or less references is not a reason for which to reject this manuscript.
References included are not just a way to justify claims, but also a way for the reader to catch up with the basics needed for reading this manuscript: they would benefit for a wider coverage of the topics. My recommendation is that the authors expand on the research groups cited even at the expense of some of the already cited references (and I know I may be shooting myself on the foot, as I could find me name in the references). Here are some suggestions

- Plate behavior in convection models. Only works with a particular code are cited. And only two main researchers (Nick Coltice and Paul Tackley). Even if I should be a bit more 'loyal' to these groups, the fact is that this issue has been a topic addressed all over the world. The authors may indeed leave some of the references cited, but they can also add Trompert and Hansen (1998), Moresi and Solomatov (1998), or any other suitable group (my suggestion is that they try balancing the continents included).
- Low temperature plasticity. It is understandable that the previous work by the authors (e.g. Garel et al., 2020) takes a prominent role. And yet, I cannot help but bark at the exclusion of some groups out there. In particular, I was surprised to see no reference to the most recent (to my knowledge, I am not an expert) experiments on low temperature plasticity (e.g. Hansen et al. 2019; Warren and Hansen, 2023; admittedly the latter is a review, please check references therein). Of course, the scope of that literature is different to that of this manuscript, but the total absence of some of this work is surprising, at the very least. I am not a rock/mineral physicist, though, and there could be some reason that eludes me for which these works are not cited.

- Inheritance. Some authors that have worked on that topic will be absent by need (the authors cannot cover everything), but again, this is a bit unbalanced. By comparison, for example Fuchs is cited three times (probably deservedly so), while too many others are absent (e.g. Foley and Bercovici 2014; Heron et al., 2016; there are many more, these are just some that have done numerical models with similar scope; and I am not objective or without conflict of interest here). I could not find any reason for this omission, although perhaps there is one and I have misunderstood the paper somehow.

In contrast, the literature review on models of more specific rift systems is quite impressive (although, IMO, still a bit continent-centric).

**Figures:**

Consider increasing labels fonts when needed/possible (see below).

Figure 9: Please clarify the axes (if parentheses mean units, they mean units, if they don't, they don't). Axes are confusing. Pa s vs. Pa.s notation differ in x-axis label in panels a and c. In these panels the parentheses seem to signal units while in panels b and d seem to signal conditions; please use brackets, parentheses, etc consistently.

**Appendix A:** I'm sure the boundary conditions are correct, but are not clear as written now. The authors mention a linear system for a non-linear rheology, a changing temperature profile, etc, which is not properly described.

- Line 594: "Incompressible Navier-Stokes equations […] for this 1-d profile simplify to:" -> change to "Incompressible Stokes equations […] for each of these layers simplify to:". Otherwise this would not be strictly correct, or/and not be able to result in $10^{th}$ order polynomial later on.

I haven't checked the Garel and Thoraval paper (sorry) but I am going to assume this 10th order polynomial poses no problem and it is going to hold with a changing thermal structure in the lithosphere, about which I need to remain skeptical. Still, please clarify a bit better this part.

**MINOR SUGGESTIONS**

Line 2: "Viscosity is capped": viscosity is not capped, technically, what is capped is the stress (note correct use in line 14). This is a minor technicality that I am sure won't confuse geodynamicists, but may be hard to understand to other geoscientists outside this field.

Line 3: "[...] have been proposed, among which low-temperature plasticity.". This construction may make sense in many languages (it does in mine), but I don't think it does in English. Usually, the subordinate clause after "among which" would need a verb. Probably just "among which low-temperature plasticity is the most discussed/interesting/promising", or something similar, is enough.

Lines 11-12. The use of respectively is peculiar. Although I don't dare to say it is wrong. Online, I could not find this use. https://www.merriam-webster.com/sentences/respectively ; https://www.internationalscienceediting.com/how-to-use-respectively-correctly/ ; https://blog.mdpi.com/2022/11/10/how-to-use-respectively/ . Please double-check this, perhaps with a native speaker (some colleagues have commented to me they have indeed seen this before). Otherwise change to "[...] is either fully fully mechanical or fully thermal for temperatures lower or higher than 1300 K, respectively." See throughout the text.

Line 26. This is true according to that specific paper. However, note that lateral strength contrasts may arise as well from chemical heterogeneities (e.g. Bodinier and Godard, 2013) which may, or not, imply weakening or strengthening; topography (Turcotte and Schubert, 2014). Of course, weakening may still be needed for creating plate boundaries, but not necessarily to sustain lateral strength contrasts. I have realized this is not important for this paper, but notice that this sentence is an assumption, not a necessity.

Lines 26-32: The authors may find useful the paper by Gerya (2024).

Line 39: These citations are very unbalanced, all of them by the same code. Consider removing some of these (not so relevant), and adding, e.g. (but not necessarily) Trompert and Hansen (1998); and/or Moresi and Solomatov (1998); and/or something by other groups (e.g. Höink and Lenardic 2010).

Line 182: I don't think the definition of plateness is needed, as the authors refer to it correctly, in case the authors wanted to summarize the manuscript.

Line 220. I'm not sure whether describing the sections in section 4.1 already makes sense. Wouldn't make sense to include a paragraph before 4.1, as an introduction after 4 before introducing the subsections?

Line 243: Note this is the use of respectively I am accustomed to and that I have seen elsewhere.

Line 245: (0-6.5 Myr *for the reference case*). This may seem redundant, but it is important to highlight that the duration of the stages is dependent on the case.

Line 250. What are plates outskirts? I tried to figure out what this is, and I think it would be better to just change "plate outskirts" to "plates".

Section 4.2: Two stages vs. three stage. The authors comment two stages, but in reality there are three (by their own definition). And regarding this...

I am a bit of a stickler for steady state. While the authors mention stationary and not steady state, I would disagree with their evaluation. The duration of the third stage is not long enough to evaluate anything, let alone its stationarity (which again, I interpreted as steady state). Rather, it seems to me that this third stage is out of the scope of this work, and some rewording may be needed.

Stage 1 and Stage 2: Add "for the reference case" inside the parentheses for the time spans. E.g. "[...] spatial focusing of the deformed zone (0/6.5 Myr for the reference case).

Line 278: Note that the yield stress approximation is one of several possibilities, and that the "need" in this line is not a hard "need" but a soft "need".

Line 332: I don't understand. Why another feedback? Why not the same non-linear feedback described before?

Line 344: Is it necessary to introduce what you are going to do in the section? I understand a small summary at the beginning of section 4 but this is a bit of redundancy on an article that is already a bit long.

Line 362,363: Are these lines needed if they are going to be commented about below?

Line 365: Without boundary conditions that are fully consistent with the thermal state of the lithosphere, this is a very strong sentence. It may be true, but the initial plate age may not alter much the duration of strain localization.

Line 372: velocity larger than what? (0.2 cm yr -1? 1 cm yr -1? Please specify). Perhaps change to "the larger the velocity the faster the plate thinning".

Line 379. Are you sure you are not opverinterpretting the data? It seems to me that the "plateau" is defined by way too few points and it looks pretty linear in logarithmic scale to me. The scatter within a single velocity value (different plate ages) may  be too much for this level of interpretation. Besides, how does this affect the overall implications or conclusions or the paper? I would consider summarizing this a next paragraphs (or perhaps even 1 or 2 before) into fewer, less interpretative ones.

Line 410. I agree, and this is useful. But why is it important to disentangle the thermal and mechanical weakening components? I think the paper would improve substantially if the authors were more specific on the why, rather than the how.

Line 415. In ". Here, it highlights ...". What does this "it" refer too?

Lines 419-421. While these references are more diverse than in the "Introduction" section, they are not greatly used. The work by Maxim does not feature any weakening that is not purely thermal, ergo does not need any disentanglement; the work of Fuchs does not include grain size and should not be used to reference a grain size field (note that, if you'd like to diversify the references, this would be a good point to include some references I mentioned in the main comments)... I am of course glad to see these groups cited, but please make sure the references are properly used.

Line 434-435. "The colder heart of the mantle lithosphere". Not very clear to me, it is definitely quite nice metaphor, but I think the readers would prefer a word a bit more specific than "heart" (note that the rest of the manuscript is quite technical and precise, this is just a very specific note).

Line 535. "@@": meaning?

Figures:

Figure 5. Is the resp. in figure 5 "respectively"? Please check use and consider whether abbreviations help or not.

Figure 7. Again, I understand the template-to-final issues, but if the figure is meant to be this small, consider increasing the labels and legends font (note differences between figure 7 and 8 legend labels).

Figure 8 is a bit dense, consider simplifying (are all panels needed to defend the conclusions of this manuscript?).

Figure 9: Axes are confusing. Pa s vs. Pa.s notation differ in x-axis label in panels a and c. In these panels the parentheses seem to signal units while in panels b and d seem to signal conditions; please use brackets, parentheses, etc consistently.

Figure 10: Ten panels may be overkill for a vertical-placed figure. Consider place it horizontally or take two panels out (which would in reality mean only taking one panel out). Re-check font sizes then.

Figures 6 and 11. It is not clear what the white areas represent. Since the red-to-green colormap is a percentual map in the legend of Figure 6, it cannot be that these areas are beyond or below the color map values (saturation). I assume it means no weakening, but this is just an assumption (please specify in the legend of Figure 6, at the very least).

Appendixes A and B: I am a bit worried about the choice of boundary conditions. Particularly for extensional settings the choice between velocity and stress boundary conditions is often a matter of debate. Moreover, the bottom boundary injection of material may influence the results considerably, because Dynamic

pressure is one of the variables for which we solve, strain rate (particularly) may be affected by the injection of material. I have no particular opinion on the right way to proceed, I am not that knowledgeable in extensional settings, but in the ideal world, the authors could show a preliminary model or test case with different boundary conditions showing that the results are not greatly affected. Nevertheless, if the authors do not find this feasible, this is not an issue that should preclude the publication of the paper.

**REFERENCES**

Bodinier J.-L. and Godard M. 2013. Orogenic, Ophiolitic, and Abyssal Peridotites. *In* Treatise of Geochemistry. https://doi.org/10.1016/B978-0-08-095975-7.00204-7

Duretz, T., de Borst, R., Yamato, P., Le Pourhiet, L., 2020. Toward Robust and Predictive Geodynamic Modeling: The Way Forward in Frictional Plasticity. Geophysical Research Letters, 47(5), e2019GL086027. https://doi.org/10.1029/2019GL086027

Duretz, T., Räss, L., de Borst, R., Hageman, T., 2023. A Comparison of Plasticity Regularization Approaches for Geodynamic Modeling. Geochemistry, Geophysics, Geosystems, 24, e2022GC010675. https://doi.org/10.1029/2022GC010675

Foley, B., and Bercovici, D. 2014. Scaling laws for convection with temperature-dependent viscosity and grain-damage. Geophysical Journal International, 199, 580-603. https://doi.org/10.1093/gji/ggu275

Garel, F., Thoraval, C., Tomassi, A., Demouchy, S., and Davies, D. R., 2020. Using thermo-mechanical models of subduction to constrain effective mantle viscosity. Earth and Planetary Science Letters 539, 116243. https://doi.org/10.1016/j.epsl.2020.116243

Gerya, T. 2024. Large-scale-long-term Strength of the Lithosphere: New Theory and Applications. Petrology 32, 128-141. https://doi.org/10.1134/S086959112401003X

Gouriet, K., Cordier, P., Garel F., Thoraval, C., Demouchy, S., Tommasi, A., and Carrez, P., 2019. Dislocation dynamics modelling of the power-law breakdown in olivine single crystals: Toward a unified creep law for the upper mantle. Earth and Planetary Science Letters, 506, 282-291. https://doi.org/10.1016/j.epsl.2018.10.049

Hansen, L. N., Kumamoteo, K. M., Thom, C. A., Wallis, D., Durham, W. B., Goldsby, D. L., Breithaupt, T., Meyers, C. D., and Kohlstedt, D. L. 2019. Low-Temperature Plasticity in Olivine: Grain Size, Strain Hardening, and the Strength of the Lithosphere. Journal of Geophysical Research: Solid Earth, 124, 5427-5449. https://doi.org/10.1029/2018JB016736

Heron, P. J., Pysklywec, R. N., and Stephenson, R. 2016. Identifying mantle lithosphere inheritance in controlling intraplate orogenesis. Journal of Geophysical Research: Solid Earth, 121(9), 6966-6987. https://doi.org/10.1002/2016JB013460

Moresi, L., and Solomatov, V., 1998. Mantle convection with a brittle lithosphere: thoughts on the global tectonic styles of the Earth and Venus. Geophysical Journal International 133(3), 669-682. https://doi.org/10.1046/j.1365-246X.1998.00521.x

Trompert, R., and Hansen, U., 1998. Mantle convection simulations with rheologies that generate plate-like behaviour. Nature 395, 686-689. https://doi.org/10.1038/27185

Turcotte, D., and Schubert, G. Geodynamics. 3rd ed. Cambridge University Press 2014.

Warren J. M., and Hansem, L. N., 2023. Ductile Deformation of the Lithospheric Mantle. Annual Review of Earth and Planetary Science, 51, 581-609. https://doi.org/10.1146/annurev-earth-031621-063756